# AutoLLMResearch: Training Research Agents for Automating LLM Experiment Configuration — Learning from Cheap, Optimizing Expensive

## Abstract

Effectively configuring scalable large language model (LLM) experiments, spanning architecture design, hyperparameter tuning, and beyond, is crucial for advancing LLM research, as poor configuration choices can waste substantial computational resources and prevent models from realizing their full potential. Prior automated methods are designed for low-cost settings where repeated trial and error is feasible, but scalable LLM experiments are too expensive for such extensive iteration. To our knowledge, no work has addressed the automation of high-cost LLM experiment configurations, leaving this problem labor-intensive and dependent on expert intuition. Motivated by this gap, we propose AutoLLMResearch, an agentic framework that mimics how human researchers learn generalizable principles from low-fidelity experiments and extrapolate to efficiently identify promising configurations in expensive LLM settings. The core challenge is how to enable an agent to learn, through interaction with a multi-fidelity experimental environment that captures the structure of the LLM configuration landscape. To achieve this, we propose a systematic framework with two key components: 1) *LLMConfig-Gym*, a multi-fidelity environment encompassing four critical LLM experiment tasks, supported by over one million GPU hours of verifiable experiment outcomes; 2) *A structured training pipeline* that formulates configuration research as a long-horizon Markov Decision Process and accordingly applies *Train/Test Experiment Curation*, *Trajectory Simulation*, *Policy Distillation* and *Multi-turn Reinforcement Learning* to incentivize cross-fidelity extrapolation reasoning. Extensive evaluation against diverse strong baselines on held-out experiments demonstrates the effectiveness, generalization, and interpretability of our framework, supporting its potential as a practical and general solution for scalable real-world LLM experiment automation.

## 1 Introduction

As Large Language Models (LLMs) are deployed across increasingly diverse scenarios, the need to tune them for different scales and settings has grown rapidly. Such tuning hinges on a series of configuration decisions, including choices of hyperparameters (Kaplan et al., 2020), architecture-related settings (Sukthanker et al., 2024), training recipes (Yang et al., 2021), and data-mixture design (Ye et al., 2025), which together shape model quality and efficiency; poor choices waste substantial compute and prevent models from realizing their full potential (Hoffmann et al., 2022; Halfon et al., 2024). Yet identifying effective configurations remains highly labor-intensive and expert-driven, especially as experiments scale up and become costly to rerun, making configuration research for scalable LLM experiments practically important and insufficiently studied. Recent Auto-research methods (Karpathy, 2026; OpenAI, 2025; Jiang et al., 2025) and existing optimizers (Akiba et al., 2019) aim to automate optimizing the configuration tuning workflow. However, they are predominantly designed for low-cost settings (classical ML such as DecisionTree, SVM, etc) where agents can propose configurations, execute them multiple times, and iterate extensively based on prior outcomes. This paradigm does not work well for large-scale LLM experiments (e.g., ≥7B models or ≥20B training tokens), where even a single training run consumes hundreds of GPU hours and only a few trials are feasible.

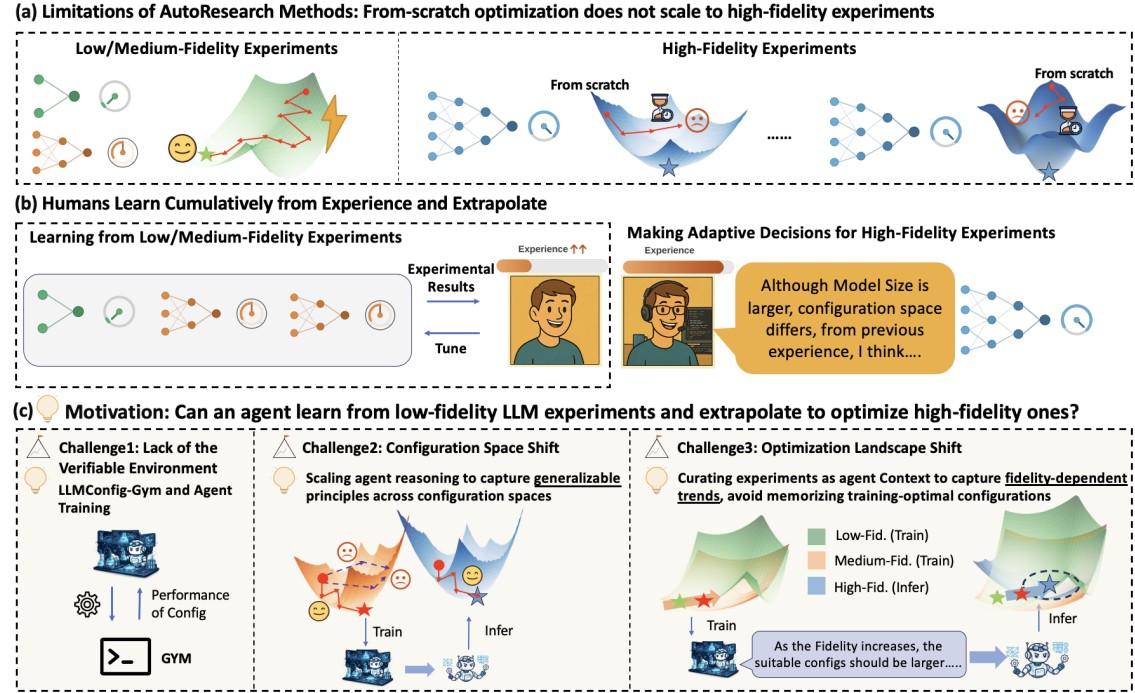

Figure 1: **Overview of the limitations of current methods, our motivation, and the three challenges** addressed by our framework.

Table 1: **Comparison of approaches.** Columns C1–C3 correspond to three challenges mentioned in Introduction. Only our **AutoLLMResearch** supports cumulative experiential learning with a verifiable LLM Experiment environment (LLMConfig-Gym), and addresses both cross-fidelity shifts.

| Category | Method | Target Experiments | Cumulative Experiential Learning | C1: Verifiable LLM Environment | C2: Space Shift | C3: Landscape Shift |
|---|---|---|---|---|---|---|
| Before LLM | Traditional HPO Tools | Low-Cost | × | × | × | × |
| | OptFormer (Chen et al., 2022) | Low-Cost | ✓ | × | × | × |
| | MetaBO (Volpp et al., 2020), NAP (Maraval et al., 2023), FSBO (Wistuba and Grabocka, 2021) | Low-Cost | ✓ | × | × | × |
| LLM-Based Prompting | LLM (GPT-5, Gemini, O4-mini, etc) (Liu et al., 2024) | Low-Cost | × | × | × | × |
| **LLM-Based RL Training** | **AutoLLMResearch (Ours)** | **High-Cost** | ✓ | ✓ | ✓ | ✓ |

These approaches, therefore, cannot converge on good settings within realistic budgets. To our knowledge, no prior work has explicitly addressed the automation of such high-cost LLM experiment configuration, leaving a significant and growing gap between the need to discover high-performing configurations under limited trials and the methods available.

Motivated by this gap, we present, to our knowledge, the first systematic study on whether, and how, expensive LLM experiment configuration can be effectively automated. We identify that the core challenge lies in finding **good configurations under strict budget constraints**, where only a handful of costly trials are feasible. To overcome this, we draw inspiration from how human researchers learn to optimize LLM experiments: they develop generalizable principles from low-fidelity (low-cost) experiments and extrapolate them to high-fidelity (high-cost) configuration settings. Some prior meta-learning works, such as (Volpp et al., 2020; Maraval et al., 2023; Wistuba and Grabocka, 2021), also emphasize cumulative experiential learning across prior experiments; however, they are designed only for same-fidelity transfer, which is considerably easier since there is no need to extrapolate across fidelities, and they struggle with the LLM-experiments-specific challenges detailed below. Our key insight is that the text-based reasoning capabilities of LLM agents can be harnessed to enable cross-fidelity extrapolation, leading to our central question:

*Can an agent learn from low-fidelity LLM experiments and extrapolate to optimize high-fidelity ones?*

Building on this motivation, we further identify three key challenges unique to this cross-fidelity learning scenario, illustrated in Fig. 1: 1) **Challenge 1: Lack of Verifiable Environment for LLM Experiments.** No existing environment provides verifiable multi-fidelity LLM experiment outcomes for enabling agents to learn from experience across fidelity levels. 2) **Challenge 2: Configuration Space Shift:** The configuration space differs between training (low-fidelity) and target (high-fidelity) experiments; the agent must reason across this shift and capture generalizable principles across them. 3) **Challenge 3: Optimization Landscape Shift:** Even within the same configuration space, the optimization landscape changes across fidelity levels, and optimal configurations do not necessarily transfer monotonically. Hence, rather than memorizing the best from training, the agent should reason about fidelity-dependent trends and adapt its decisions accordingly. In light of these three challenges, as summarized in Table 1: HPO tools (Akiba et al., 2019; Scikit-Optimize) and LLM-based auto-research methods are all designed for low-cost experiments. Without leveraging cumulative experimental experience, they optimize each individual experiment from scratch, making them poorly suited to high-fidelity settings where even a single trial can be extremely expensive. Meta-training methods do support experiential learning from prior experiments, but only target the same-fidelity small-scale machine learning tasks, where the learned knowledge is encoded as fixed probability distributions over a fixed configuration space, making them prone to overfitting and unable to address Challenges 2 and 3.

To address all three challenges, we observe that LLMs inherently can accumulate experience through training and operate on text (supporting flexible configuration-space). Under RL reward signals in agentic training, there is potential to train an LLM agent to reason like a researcher: learning from prior low-fidelity experiments and extrapolating to high-fidelity decisions. Building on this idea, we propose a systematic framework with two key components: 1) **LLMConfig-Gym**, a multi-fidelity environment encompassing four critical LLM experiment tasks. It serves as an interactive environment that supplies pre-computed experiment results to construct verifiable rewards for each configuration the agent proposes, enabling end-to-end multi-turn RL training. 2) **A structured training pipeline** that formulates the configuration problem as a long-horizon Markov Decision Process (MDP), where the agent reasons over prior observations and proposes new configurations within LLMConfig-Gym. The pipeline combines Trajectory Simulation, Policy Distillation, and Multi-turn RL to incentivize researcher-like extrapolation from cheap experiments to expensive ones, e.g., extrapolating from ≤3B / 10B-token experiments to 7B / 20B-token ones. More broadly, our work represents a concrete step toward Recursive Self-Improvement (RSI) (Good, 1966; Schmidhuber, 2006; Zhuge et al., 2026; Rank et al., 2026; Zhang et al., 2026; Wang et al., 2026): rather than relying on heuristics, we train an Agent that learns to extrapolate from cheap experiments and automate the very process of training AI, a capability whose value grows as experiment costs scale up. In summary, our contributions are:

- To our knowledge, this is the first systematic study on automating expensive LLM experiment configuration. Our central idea is to train an LLM-based agent that accumulates knowledge from low-fidelity experiments and extrapolates it to guide high-fidelity decisions.

- We design LLMConfig-Gym and a training pipeline that together enable an LLM-based agent to conduct cumulative experiential learning, and achieves strong cross-fidelity extrapolation.

- Extensive quantitative and qualitative experiments on four representative LLM configuration tasks (Model Architecture, Pretraining Hyperparameter, RL GRPO Tuning, Data Mixture) across models up to 7B or training tokens up to 20B demonstrate the superior performance of our approach, and in-depth analysis confirms the generalization and interpretability of the trained agent, providing natural-language explanations about its cross-fidelity reasoning process.

## 2  Related Work

**Methods for Automating LLM Experiment Configuration:**  Since no prior work addresses experience transfer across fidelity levels, we organize adjacent methods into: **1) Meta-Bayesian optimization:** MetaBO (Volpp et al., 2020) and NAP (Maraval et al., 2023) learn a meta-probabilistic model over configurations from offline experiments to guide optimization on test problems. However, they operate exclusively in same-fidelity settings and lack the ability to extrapolate across different fidelity levels and configuration spaces. **2) HPO tools and LLM-based AutoResearch methods:** Classical HPO tools optimize on-policy

within a fixed configuration space and cannot handle the configuration-space shift. Recent LLM-based work (Liu et al., 2024; Zhang et al., 2023; Mahammadli and Ertekin, 2025; Liu et al., 2025) uses LLMs as optimizers, proposing and refining configurations based on textual descriptions and past results. While LLM-based approaches handle diverse configuration spaces, both rely on on-policy methods that assume many experiment executions, an assumption that is infeasible for high-cost LLM experiments where each run is prohibitively expensive. *Unlike all the above, we are the first to train a text-reasoning agent that learns transferable principles from low-fidelity experiments and extrapolates them to high-fidelity LLM configuration decisions.*

**Incentivizing LLM Reasoning by RL with Verifiable Rewards:** Reinforcement Learning with Verifiable Rewards (RLVR) has advanced generalization and reasoning capabilities of frontier LLMs (OpenAI, 2024; Guo et al., 2025; Team et al., 2025). *Building on these advances, we are the first to construct an RL Gym-style environment for LLM experiment configuration and leverage RLVR to incentivize an LLM agent to reason like a researcher, yielding more sample-efficient and interpretable configuration decisions.*

## 3   LLMConfig-Gym: Environment for Training Agents

We build the first gym for training and evaluating agents on LLM experiment configuration. Here, we briefly introduce design principles (tasks, organization, interface), and leave more in Appendix A.

Table 2: **Task characteristics of four cross-fidelity LLM experiment configuration tasks.** Each task is categorized by its primary challenge: configuration space shift (Tasks 1 & 4) or optimization landscape shift (Tasks 2 & 3), along with the corresponding training and testing configuration spaces.

| Task | Opt. Target | Primary Challenge | Training Space (Low/Med Fidelity) | Testing Space (High Fidelity) |
|---|---|---|---|---|
| Task 1: Model Architecture Design | Balance Perplexity & Latency | Configuration Space Shift | *GPT-M:* embed$\in \{256, 512, 1024\}$, layers$\in \{22, 23, 24\}$, per-layer: heads$\in \{8, 12, 16\}$, mlp_ratio$\in \{2, 3, 4\}$, bias$\in \{T, F\}$ | *GPT-L:* embed$\in \{320, 640, 1280\}$, layers$\in \{34, 35, 36\}$, per-layer: heads$\in \{8, 16, 20\}$, mlp_ratio$\in \{2, 3, 4\}$, bias$\in \{T, F\}$ |
| Task 2: Pretraining Hyperparameter | Loss $\downarrow$ | Optimization Landscape Shift | Learning_Rate$\in \{2^{-10.5}, \ldots, 2^{-7.0}\}$, Batch_Size$\in \{32K, \ldots, 4M\}$ *Train:* training tokens$\leq$ 8B, model params$\geq$ 429M | *Test:* training tokens$\geq$ 20B, model params$\geq$ 214M |
| Task 3: RL GRPO Tuning | Reward $\uparrow$ | Optimization Landscape Shift | Learning_Rate$\in \{10^{-6}, 5 \times 10^{-6}, 10^{-5}\}$, Batch_Size$\in \{16, 32, 64\}$, $\lambda_{KL} \in \{0, 10^{-3}\}$ *Train:* MMLU subsets, 256 samples | *Test:* GSM8K/DAPO, 768–1536 samples |
| Task 4: Data Mixture for Instruction-Tuning | Test Performance $\uparrow$ | Configuration Space Shift | *Different data mixture ratio space between training (Qwen2.5-3B) and testing (Qwen2.5-7B); details in Section A.6* | |

**Desired Agent Capabilities, Task Design and Hierarchical Organization:** A central goal of our framework is to enable cumulative experiential learning, which requires a well-defined offline environment that does not yet exist. To fill this gap, we built LLMConfig-Gym for RL training and evaluation. After surveying the key design choices in LLM experiments, we identify four representative tasks as shown in Table 2: 1) **Model Architectures** such as the number of transformer layers, embedding dimensions, and attention heads that directly influence the trade-off between model perplexity and latency; 2) **Pretraining Hyperparameters** such as learning rate and batch size that significantly affect pretraining loss; 3) **RL GRPO Tuning Hyperparameters** including the choices of learning rate, batch size, and KL loss coefficient that govern the reward achieved during GRPO tuning of a base LLM; and 4) **Data Mixture Weight Ratios**, which play an important role in model performance and arise frequently in practice. LLMConfig-Gym adopts a hierarchical structure organized as **Task → Fidelity → Experiment**. For each task, we collect experiments at multiple fidelity levels by leveraging open-source datasets such as HW-GPT-Bench (Sukthanker et al., 2024) or running grid-search experiments offline. To preserve flexibility, we deliberately do not impose a rigid fidelity definition. Instead, we expose fidelity-related metadata (model size/training tokens/etc), enabling flexible usage.

**Unified Format, Interface and Sufficient Text-Based Context:** Since LLM experiment runs are too costly for online interaction, we unify all tasks into an offline **Lookup Table** built from massive offline runs. The core API is a *tell* function that takes a configuration and returns its performance and experimental details (e.g., loss) **within seconds**. To our knowledge, no prior work offers such a systematic, ready-to-use gym for LLM experiment configuration; we open-source LLMConfig-Gym as a **fast, broadly reusable** gym that any researcher can plug in to train or evaluate new methods, lowering the barrier to entry for automated

LLM experiment research. A natural advantage of building agents on LLM is the native compatibility with rich text. We therefore design per-task metadata (task/configuration/etc descriptions) to help the agent interpret each problem and make informed decisions. [1] Metadata details are in Appendix A.

## 4 Methodology of AutoLLMResearch

### 4.1 Problem Formulation: MDP for Agentic Training

We formulate LLM configuration as a sequential optimization process where the LLM-based agent serves as the policy, reasoning via text to make configuration decisions.

- **Environment:** Our constructed LLMConfig-Gym. The gym provides a *tell* function that receives a configuration $x$ and returns its performance $y$ and experimental details.

- **Policy and Action space:** The policy $\pi_\theta(a_k \mid s_k)$ is parameterized by an LLM and optimized end-to-end via RL. The action space $a_k \in \mathcal{A}$ consists of two text-based steps per trial: (Think): given the context, the agent reasons step by step to identify a promising configuration; (Execute): it commits the chosen configuration to the Gym to observe its performance.

- **Observation:** Performance and additional experimental details, returned as text by the Gym.

- **State and Transition:** The state is defined as $s_t = (H_t, t, T)$, with $H_t = \{\langle x_1, y_1\rangle, \ldots, \langle x_{t-1}, y_{t-1}\rangle\}$ the history up to step $t$, and $T$ the total budget. The transition appends the latest evaluation: $H_{t+1} = H_t \cup \{\langle x_t, y_t\rangle\}$ and increment $t$.

- **Objective:** Maximize expected reward under a budget: $\arg\max_{\pi_\theta} \mathbb{E}\left[R\left(\{y_1, \ldots, y_T\}\right)\right]$, with each turn $s_{t-1} \xrightarrow{\text{(Think)}} x_t \xrightarrow{\text{(Execute)}} \text{GYM} \rightarrow y_t$ for $t = 1, \ldots, T$, optimized via multi-turn RL.

(Think) **as Researcher-Level Reasoning.** A key distinction of our formulation is that optimizing $\pi_\theta$ directly shapes (Think), which operates in long-form text-based reasoning. We find this lets the agent analyze concrete experiments alongside fidelity information step by step, yielding stronger extrapolation. This differs fundamentally from prior meta-training methods which model same-fidelity learning as a categorical distribution $p(x_i \mid s_k)$ over a fixed configuration set $\{x_1, \ldots, x_n\}$ and merely select $\arg\max_{x_i} p(x_i \mid s_k)$, whereas our text-based (Think) encourages the agent to internalize the reasoning steps that lead to better configurations rather than memorize a fixed distribution.

### 4.2 End-to-End Training Agent in LLMConfig-Gym

#### 4.2.1 Step1: Train/Test Experiment Curation

This step builds training and testing samples from Gym experiments addressing Challenge 1.

**Addressing Challenge 2 via Rich Text Information throughout Input and Rollout.** Recall Challenge 2: configuration spaces shift across fidelities; our goal is to scale agent reasoning to capture generalizable principles. Our idea is to *build rich textual context throughout, leveraging the LLM's pretrained domain knowledge to fully understand the problem and reason effectively.* As shown in the input frame of Fig. 2, the agent's input has three modules: **Task** (task description and optimization target); **Context** (fidelity information, in-context demonstrations, configuration space, budget); **Instructions** (guidance). Including budget lets the agent adapt its exploration–exploitation trade-off to the remaining turns, which is critical for high-cost LLM experiments. We also enrich rollout feedback: after each Gym interaction, the agent receives: 1) target performance and 2) task-specific experiment details (e.g., critic scores in RL tuning). This richer grounding makes overlapping cross-fidelity patterns more visible and amplifies low-to-high fidelity transfer.

---

[1] To prevent data leakage from the LLM's pretraining corpus, we exclude dataset names.

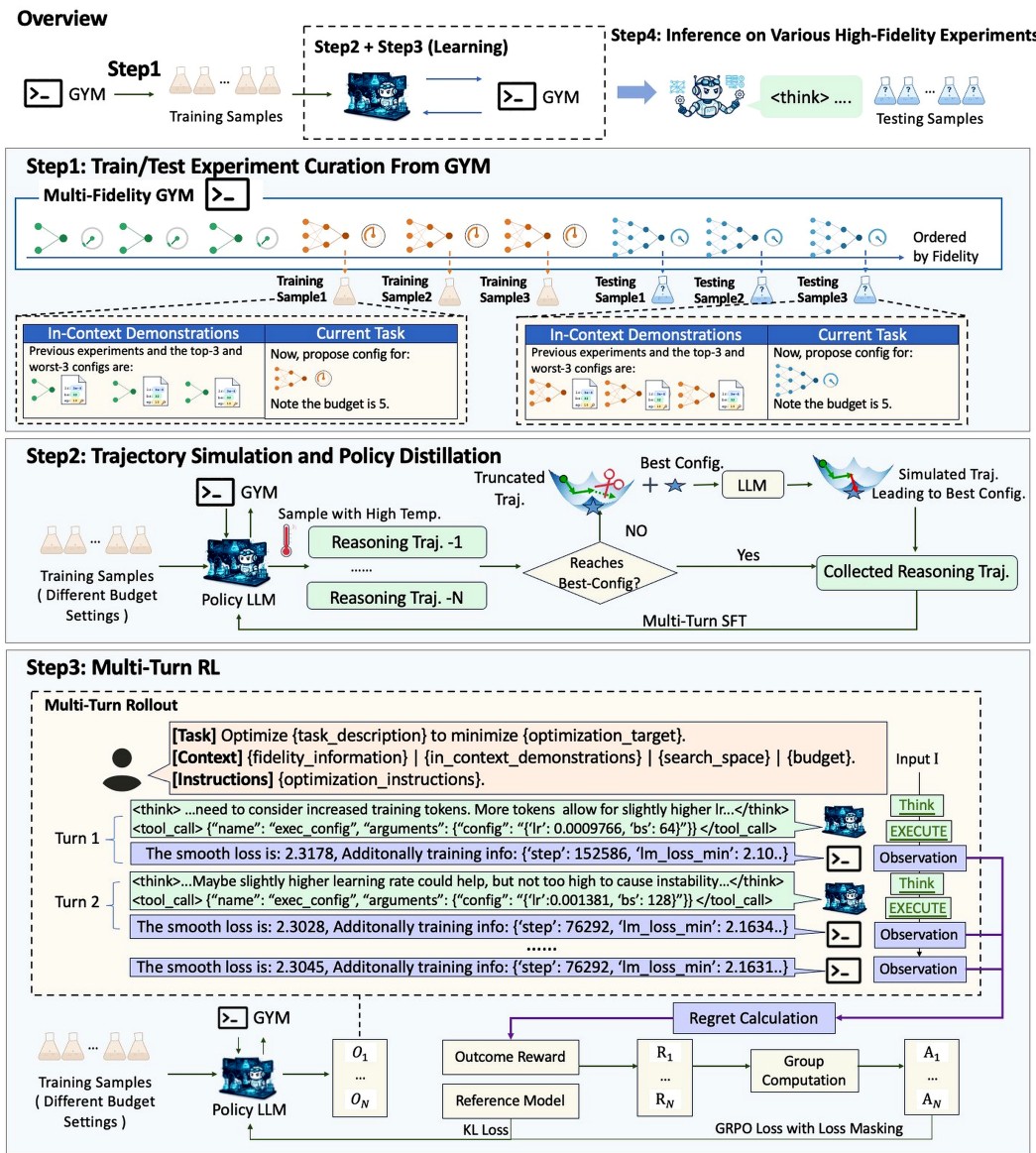

Figure 2: **Overview of our framework**. Step 1 curates multi-fidelity training and testing experiments with in-context demonstrations. Step 2 collects successful reasoning trajectories via high-temperature sampling for policy distillation. Step 3 further optimizes the policy with multi-turn RL. Step 4 deploys the trained policy on various unseen high-fidelity experiments.

**Addressing Challenge 3 via Lower-Fidelity Experiments as In-Context Demonstrations.** Recall Challenge 3: We want the agent to extrapolate across fidelities by capturing configuration trends rather than overfitting to training-time optima. Our idea is to *instruct the agent to reason about how configurations should change as fidelity increases*, by giving it lower-fidelity results with their fidelity information and asking it to analyze the trend and propose configurations for the current level. We order experiments by fidelity using domain knowledge (e.g., model size, dataset size, epochs) and split them into low-/medium-/high-fidelity sets $L$, $M$, $H$, then construct one-to-many pairs as samples: for training, each $m_i \in M$ is paired with all $L$ as in-context demonstrations; for testing, each $h_i \in H$ is paired with $M$. For each pair, Top-K configurations from the lower-fidelity side are concatenated with fidelity information in the prompt. The input for agent is thus $(L, m_i)$ during training and $(M, h_i)$ during testing. This differs fundamentally from prior meta-training methods that treat each experiment as an independent sample ($m_i$ or $h_i$ alone). *By*

*associating experiments across fidelities, our strategy puts the agent in a cross-fidelity transfer setting rather than independent exploration, encouraging transferable reasoning across fidelities.*

### 4.2.2 Step2: Trajectory Simulation and Policy Distillation

**Drawbacks of Direct RL Training.** We initially trained directly on LLMConfig-Gym, but observed two drawbacks: 1) the agent often converges to local optima, as rollouts remain in local regions without explicit supervision toward the best configuration; 2) since rollouts involve long-horizon reasoning and environment interaction, the base agent forgets instructions and produces format errors.

**Our Solution:** For each curated training sample, we augment with different budget settings to simulate budget-constrained tuning, then sample 20 rollout trajectories at temperature 0.8. Trajectories that reach the best configuration are added to a trajectory set $PD$. For samples whose trajectories all fail (i.e., stuck at local minima), we apply **Trajectory Simulation**: take the trajectory with the local-best configuration, randomly truncate the last or second-to-last trial, and prompt the LLM with 1) the truncated trajectory, 2) the best configuration, and 3) instructions to continue toward the best configuration. The truncated prefix and newly generated suffix are concatenated into a complete trajectory and added to $PD$. Finally, we perform **Policy Distillation** on the base LLM via multi-turn SFT on $PD$, applying loss masking on Gym observations and instruction tokens so the agent learns *how to reason and interact with the environment over long horizons to reach the best configuration.*

### 4.2.3 Step3: End-to-End Multi-Turn Reinforcement Learning

After Policy Distillation, we apply Multi-Turn RL via GRPO (Shao et al., 2024). For each configuration task we sample $G$ trajectories $\{O_i\}_{i=1}^G$, where $O_i = (I, a_1, \hat{y}_1, \text{obs}_1, \ldots, a_n, \hat{y}_n)$. Each trajectory receives a scalar reward $r_i$; letting $\mathbf{r} = (r_1, \ldots, r_G)$, we compute a group-normalized advantage shared by all tokens of trajectory $i$: $A_{i,t} = (r_i - \text{mean}(\mathbf{r}))/(\text{std}(\mathbf{r}) + \varepsilon)$, $\forall t$. We apply loss masking to experiment observations and instruction tokens so the agent focuses on learning the thinking process (Think and Execute). The resulting GRPO objective is $\mathcal{J}_{\text{GRPO}}(\theta) = \mathbb{E}\big[\frac{1}{G}\sum_{i=1}^G \frac{1}{|\mathcal{M}_i|} \sum_{t \in \mathcal{M}_i} \min(r_{i,t}A_{i,t}, \text{clip}(r_{i,t}, 1-\epsilon, 1+\epsilon)A_{i,t}) - \beta\,\mathbb{D}_{\text{KL}}[\pi_\theta \,\|\, \pi_{\text{ref}}]\big]$, where $r_{i,t} = \pi_\theta(o_{i,t} \mid q, o_{i,<t})/\pi_{\theta_{\text{old}}}(o_{i,t} \mid q, o_{i,<t})$ is the per-token importance ratio, $A_{i,t}$ is the token-level advantage, and the mask $\mathcal{M}_i$ retains only thinking-related tokens.

**Regret-Based Outcome Reward.** We aim to teach the agent extrapolative reasoning. To this end, we design **cumulative regret**, which scores behavior across all $T$ turns rather than only the best-found configuration, reducing overfitting. Given the distinct configurations and their performances $\{y_1, \ldots, y_T\}$, we normalize the gap between the cumulative performance $\sum_{t=1}^T y_t$ and the upper bound $T \cdot y_{\text{best}}$ by the worst-case range:

$$R_{\text{outcome}} = \begin{cases} -\dfrac{T \cdot y_{\text{best}} - \sum_{t=1}^T y_t}{T \cdot y_{\text{best}} - T \cdot y_{\text{worst}}}, & \text{if the agent proposes } T \text{ distinct valid configurations,} \\ -1, & \text{otherwise (i.e., on repeats or invalid outputs),} \end{cases} \tag{1}$$

where $y_{\text{best}}$ and $y_{\text{worst}}$ are the best and worst task performances. This design has two benefits: 1) summing over $T$ distinct trials rewards consistently high-quality proposals throughout the budget, rather than stumbling on a single good one, which is prone to overfitting in cross-fidelity transfer; 2) the worst reward $(-1)$ on repeated or invalid outputs explicitly penalizes trivial replay and format errors, encouraging genuine exploration and reliable output under budget constraints.

**Make Reward Dense via Most-Similar Configuration Matching.** With above reward, we observe that on some tasks most rewards collapse to $-1$, producing a sparse signal. The root cause is **format violations in long-horizon reasoning**: despite explicit Gym-call instructions, the agent occasionally produces minor format errors in later turns, especially for long configurations. For instance, when specifying per-layer head counts in a 4-layer Transformer, it may output `[1,1,2,3)` or `[1,1,2,3,3]` instead of a valid 4-element array. Such errors mark entire rollouts invalid and starve reward. To mitigate this, we add a **most-similar configuration matching** during Gym queries: if the generated configuration lies in valid space, we use it

directly; otherwise, we use the valid configuration with the longest common substring (Foundation, 2026) as the query. This redirects minor format violations to the nearest valid configuration, substantially reducing sparse-reward cases and stabilizing training.

## 5 Experiments

We conduct extensive experiments to address the following research questions (RQs):

- **RQ1 (Effectiveness and Generalization):** Does training agents improve performance on LLM experiment configuration, and how well does it generalize across different scenarios?
- **RQ2 (Interpretability):** What transferable reasoning principles do agents learn from low-fidelity experiments, and how do they benefit high-fidelity ones?
- **RQ3 (Training Dynamics):** How do our training strategies stabilize agent learning?

**Benchmarks and Baselines.** We evaluate on all four LLMConfig-Gym tasks (Table 2). For each task, we design the scenario in which the agent is trained on low-/medium-fidelity experiments and tested on high-fidelity ones. To examine effectiveness under varying resources, we further adopt a **budget-constrained** protocol with budgets ranging from 1 to 5 per task. Since LLM experiment configuration problem is novel, no prior baseline covers all four tasks. We adopt representative baselines from three categories: **(1) Normal Baselines:** *Random Search* (uniform sampling) and *Top-K Warm-Start (WS)* (reusing the top-$K$ training configurations).[2] **(2) Meta-Training Methods:** NAP (Maraval et al., 2023), MetaBO (Volpp et al., 2020), and FSBO (Wistuba and Grabocka, 2021), which perform end-to-end meta-optimization with offline training, not expected to extrapolate across fidelities. **(3) LLM-Based Methods:** Strong reasoning models (OpenAI O4-mini (OpenAI, 2025), Gemini (Comanici et al., 2025), GPT-5 (Singh et al., 2025)) under the AgentHPO (Liu et al., 2025) prompting framework, representing prompt-based LLM approaches.

**Implementation Details.** 1) **LLMConfig-Gym:** contains 4 representative LLM configuration tasks, with >1M GPU hours of experiment data, multiple fidelity levels, and a unified sub-second querying API. Details in Appendix A. 2) **Training Agent:** We use Qwen3-1.7B and Qwen3-4B (Yang et al., 2025) as backbones. All training runs on a single node with 4 NVIDIA A100 GPUs (80GB). For Policy Distillation, we run LlamaFactory (Zheng et al., 2024a) with DeepSpeed ZeRO-3 offload (Aminabadi et al., 2022) and gradient checkpointing, performing full-parameter fine-tuning at learning rate 5e−6 with a cosine scheduler and per-device batch size 2. For end-to-end Multi-Turn GRPO, we use Verl (Sheng et al., 2025) with FSDP offloading and SGLang (Zheng et al., 2024b) as the inference engine, with training batch size 64, learning rate 1e−6, max prompt length 8500, max response length 13000, max agent–Gym interactions per episode 5, and rollout count 5. The agent–Gym interaction is implemented as a tool call (`exec_config`) via the verl function calling mechanism. Full prompts and hyperparameter dumps are in Appendix B.

**Evaluation Metrics.** We use normalized regret as a unified metric for comparison across tasks: Regret $= \max_{h \in \mathcal{H}_T} \frac{y^*_{\text{best}} - f(h)}{y^*_{\text{best}} - y^*_{\text{worst}}}$, where $\mathcal{H}_T$ is the set of configurations proposed under budget $T$, $f(h)$ is its performance, and $y^*_{\text{best}}, y^*_{\text{worst}}$ are the best and worst performance scores across all methods. This measures how close the best configuration found is to the global optimum (in $[0, 1]$); lower is better.

### 5.1 RQ1: Effectiveness and Generalization

#### 5.1.1 Overall Performance

Overall performance is shown in Fig. 3; per-task are in Section C.1. **Result 1:** Our approach achieves the lowest regret across all tasks and budget settings. Despite using small backbone models, our trained agents outperform substantially other baselines. Together with the ablation studies below, these results show that low-fidelity cumulative learning enables effective high-fidelity optimization.

---

[2]Since Top-$K$ WS cannot handle configuration-space shifts, for Tasks 1 and 4 we substitute Random Search for Top-$K$ WS when computing average overall performance.

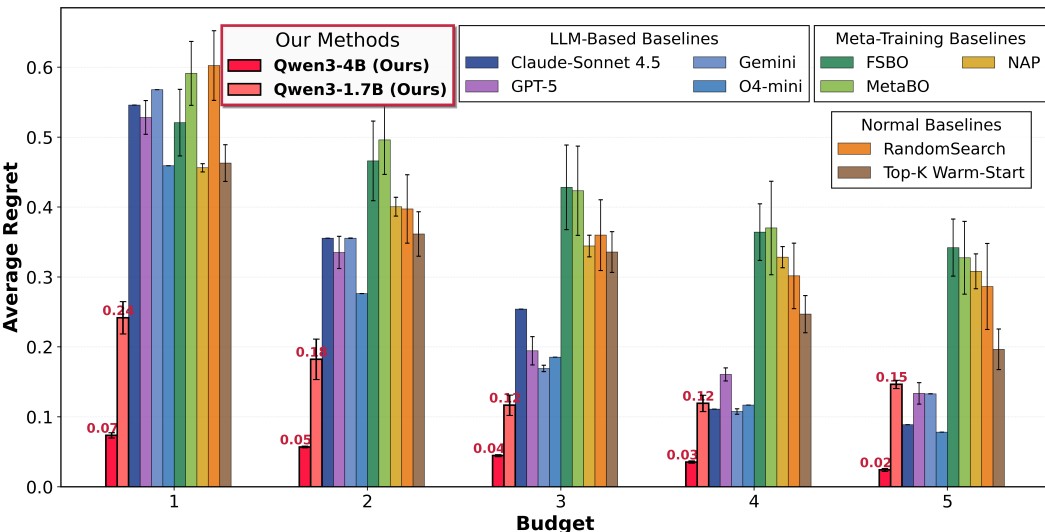

Figure 3: **Overall performance comparison across all tasks and budget constraints**. Our method achieves the lowest regret across different settings, demonstrating its effectiveness.

**Performance across Different Low-to-High Fidelity Scenarios: (1) Challenge 2:** As summarized in Table 2, Tasks 1 and 4 instantiate configuration space shift, where the agent is trained on one discretization and tested on a disjoint one. Results in Fig. 13 and 16 show that our Qwen3-4B agent reaches near-zero regret (∼0.01) from budget 2 onward on Task 1, with a similar trend on Task 4, while all other methods remain above 0.2 regret. These results confirm that Pre-LLM meta-training methods collapse on disjoint spaces because they encode fixed distributions over fixed configurations, whereas our agent transfers learned generalizable principles. **(2) Challenge 3:** Tasks 2 and 3 instantiate optimization landscape shift, where the optimal region moves between fidelities. Fig. 14 and 15 show our method achieves the lowest regret on both Tasks 2 and 3, outperforming meta-training (>0.3) and LLM-based baselines (>0.25). Top-$K$ WS, which directly reuses training-set configurations, underperforms our method, confirming that the landscape has shifted and simple configuration transfer is insufficient. **Result 2:** Across all tasks and shift types, our method consistently achieves the lowest regret, demonstrating strong cross-fidelity generalization via text-based reasoning. **Stability:** Averaged over 3 runs, it shows consistently smaller error bars (Fig. 3), supporting reliable practical LLM experiment configuration.

### 5.1.2 Ablation Studies

**Result 3:** As shown in Table 3, every component of our method is essential for incentivizing low-to-high fidelity learning. **(a) Text-Based Information.** Relative to Qwen3-4B-Base (0.247), removing **Train/Test Experiment Curation** (asking the agent to optimize each experiment individually as in prior meta-training) raises regret to 0.302, and removing **Rich Text Information** from prompt raises it to 0.281, confirming that curation enables cross-fidelity extrapolation with a strong starting point and rich text grounds the agent's reasoning in task-specific domain knowledge. **(b) Training Pipeline.** Policy Distillation alone (0.144) teaches the reasoning process but does not directly optimize regret; Multi-Turn RL alone (0.190) yields much higher regret because under long-horizon reasoning (up to 13,000 response tokens) the agent loses track or converges to local optima without distillation's supervision. Combining both reaches **0.035**: distillation anchors instruction following throughout long-horizon, while RL with regret as the reward drives the agent toward lower-regret configuration.

### 5.1.3 Evaluation of Long-Horizon Instruction Following Capabilities

Beyond reasoning quality, our agent also requires strong **long-horizon instruction following**. Each episode iterates think→propose→Gym call under strict format constraints. We evaluate **Execution Rate** (fraction

Table 3: **Ablation studies** on different components of our method.

| Method | Avg. Regret ↓ |
|---|---|
| *Ablations Text-Based Information on Base Model* | |
| w/o Train/Test Experiment Curation | 0.302 |
| w/o Rich Text Information | 0.281 |
| Qwen3-4B-Base | 0.247 |
| *Ablations Training Pipeline* | |
| + Policy Distillation | 0.144 |
| + Multi-Turn RL | 0.190 |
| **+ Policy Distillation + Multi-Turn RL** | **0.035** |

Table 4: **Instruction-following capabilities evaluation across training methods. Execution Rate** and **Unique Config. Rate** measures the ratios of successful function calls and distinct configurations, respectively, to the allocated budget.

| Method | Exec. Rate (%) ↑ | Unique Cfg. (%) ↑ |
|---|---|---|
| Base | 82.3 | 80.2 |
| DirectRL | 97.9 | 94.7 |
| Policy Distillation | 97.0 | 96.1 |
| Policy Distillation + RL Training | **98.6** | **98.2** |

of successful Gym calls over the budget, capturing well-formatted tool calls); and **Unique Configuration Rate** (fraction of distinct proposed configurations over the budget, capturing whether it avoids redundant submissions). **Result 4:** As shown in Table 4: 1) the base model performs poorly, often producing malformed function calls; 2) DirectRL reaches a high execution rate but poor unique-config rate, as long reasoning chains cause the model to lose track and RL rewards saturate at repetitive trajectories, limiting overall performance; 3) Policy Distillation improves both, and adding RL further boosts them, demonstrating the long-horizon capability of our full method.

### 5.1.4 Amortized Cost Analysis under Scalable Deployment

Unlike prior methods that optimize each task from scratch, our method accumulates experience from low-/medium-fidelity experiments and generalizes to new high-fidelity tasks. This is tailored for the **cumulative experiential learning** setting: as more tasks are configured, the upfront meta-learning investment is increasingly amortized. To quantify this, we compare cumulative GPU cost against LLM-based prompting and Optuna as the number of test tasks scales from 1 to 30 (full estimation methodology in Appendix C.4). As shown in Fig. 4, baselines incur linear cost growth since they restart on every task, whereas our method has a one-time upfront cost with negligible marginal cost per additional task, yielding a 3.6× reduction at 30 tasks. In practice, the upfront cost can be further absorbed by idle GPU time, since low-fidelity training experiments can be scheduled

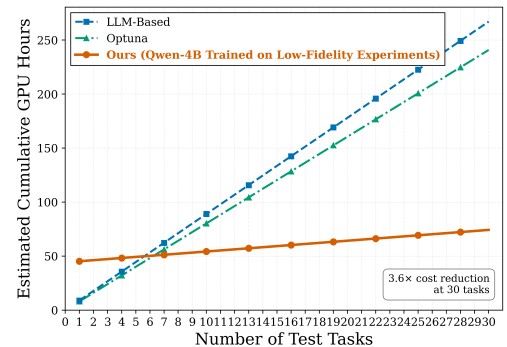

Figure 4: **Estimated cost-effectiveness under scalable deployment.**

whenever spare compute is available. **Result 5:** Our method amortizes its one-time upfront cost rapidly under scalable deployment, yielding a 3.6× cumulative GPU-hour reduction at 30 tasks compared to from-scratch baselines.

## 5.2 RQ2: Interpretations of AutoLLMResearch Agent Capabilities

### 5.2.1 From Reasoning Perspective: What Transfers from Low-to-High Fidelity Experiments

Beyond quantitative performance, we further ask a deeper scientific question: *what exactly does the agent learn that enables low-to-high fidelity extrapolation?* To answer this, we trace the agent's text-based reasoning trajectories on **both the training and test sets** across three stages: **Before Training**, **During Training**, and **Inference**. This contrast tests whether the RL-optimized reasoning captures transferable principles rather than memorizing fixed configurations, directly addressing the two challenges from the introduction. **Case Study 1** (Fig. 17, in Appendix C.2) targets Challenge 2 (Configuration Space Shift) on the Model Architecture Configuration task, and **Case Study 2** (Fig. 18, in Appendix C.2) targets Challenge 3

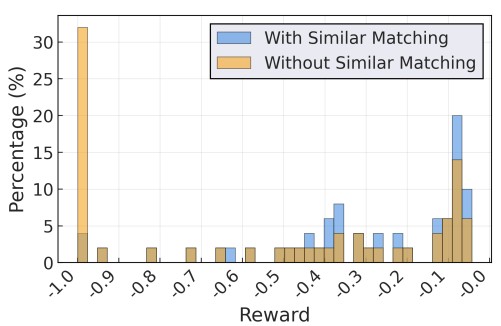

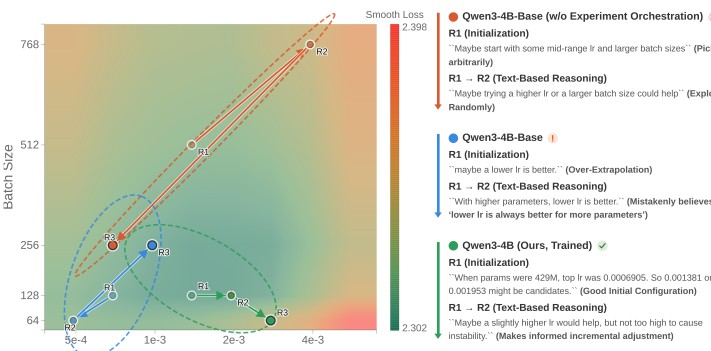

Figure 5: **Reward distribution w/ and w/o Most-Similar Matching.** W/o matching, 32% of rollouts collapse to $-1$ due to format violations. W/ matching, they are redirected to valid configurations, densifying the reward signal.

Figure 6: **Optimization trajectories of three methods on Pretraining Hyperparameter Configuration (Task 2), overlaid on the ground-truth loss landscape (darker green = lower loss).** Arrows trace each method's Round $1{\to}2{\to}3$ trajectory; ellipses mark its *search region* across runs; right-side snippets show each method's reasoning.

(Optimization Landscape Shift) on the Pretraining Hyperparameter Configuration task. **Result 6:** Across both case studies, a consistent picture emerges: **RL optimization on text-based reasoning produces transferable principles rather than memorizing fixed configurations**. In Case Study 1, the agent learns a structural balancing rule that generalizes across shifted configuration spaces. In Case Study 2, it learns fidelity-dependent trends that generalize across shifted optimization landscapes. These findings reflect what is, to our knowledge, a **fundamental and first-of-its-kind shift** in how agents learn to extrapolate for experiment configuration: training an agent to scale text-based reasoning to learn transferable principles that extrapolate from cheap to expensive experiments.

**Case Study 1: Configuration Space Shift (Model Architecture).** The training and test spaces are disjoint (e.g., `embed_dim` shifts from $\{256, 512, 1024\}$ to $\{320, 640, 1280\}$), so training-optimal configurations do not even exist at test time. The base agent defaults to the minimum `embed_dim` on both sets, whereas the RL-trained agent learns a *"balance extremes"* principle (moderate `embed_dim` with diverse architecture) and applies it to the shifted test space, yielding a $\sim$30% relative regret reduction, showing that what transfers across the configuration-space shift is not a configuration but a *scaling-aware balancing rule*. Full reasoning trajectories are in Appendix C.2.

**Case Study 2: Optimization Landscape Shift (Pretraining Hyperparameter).** Training and testing share variables (`lr`, `bs`) but differ in fidelity (2B vs. 20B tokens). The base agent latches onto an oversimplified "more tokens $\Rightarrow$ higher lr" heuristic and over-extrapolates to `lr`=5.5e-3 for a 536M-parameter model. The RL-trained agent instead learns *fidelity-dependent trends* (params $\uparrow\Rightarrow$ lr $\downarrow$; tokens $\uparrow\Rightarrow$ slightly higher lr) and selects a calibrated `lr`=1.95e-3, `bs`=128, showing that what transfers across the optimization-landscape shift is a *fidelity-dependent scaling trend*. Full reasoning trajectories are in Appendix C.2.

### 5.2.2 From Optimization Perspective: Text-Based Reasoning Prunes Search Space

Beyond reasoning analysis, we examine *whether the agent's improved reasoning translates into measurable optimization effectiveness*. We compare three variants on the same task, plotting each method's optimization trajectory and effective search region across repeated runs. Full experimental setup and trajectory details are in Appendix C.3. As shown in Fig. 6, three distinct behaviors emerge: **(1) Without Experiment Curation**, the agent explores erratically with the largest search region. **(2) The base model without training** references low-fidelity trends but extrapolates in the wrong direction, drifting toward overly conservative settings. **(3) Our trained agent** correctly extrapolates from low-fidelity experience, lands near the global optimum from R1, and concentrates its search region tightly around the optimum. **Result 7:** Text-based

reasoning learned from experience acts as a *search prior*: it provides a strong initial configuration and enables the correct trend that prunes the search space.

### 5.3 RQ3: Training Dynamics

**Quantitative Analysis.** We partition training samples by task type and track **Regret Mean@3**, **Critic Score Mean**, **Critic Score Min**, and **Response Length Mean** throughout training (Fig. 19 in Appendix C.5; metric definitions and full per-metric analysis are also there). **Regret Mean@3** decreases consistently and **Critic Score Mean** rises steadily across all four task categories. **Result 8:** Our training progressively improves configuration quality, reduces invalid outputs, and yields task-adaptive reasoning. The consistent test-set regret decrease alongside steady training reward improvement provides strong evidence that the agent learns genuinely transferable principles rather than overfitting to training configurations.

**Most-Similar Configuration Matching.** We further evaluate the effect of Most-Similar Configuration Matching on densifying the reward signal by comparing reward distributions with and without matching. As shown in Fig. 5, without matching, about 32% of samples receive the worst reward ($-1$) due to format violations. With matching enabled, this fraction drops dramatically, and the distribution shifts substantially toward higher rewards. **Result 9:** Most-Similar Configuration Matching converts otherwise wasted rollouts into meaningful training signals, stabilizing RL training.

## 6 Discussion, Broader Impact and Future Work

**Stress-Testing the Boundary of Low-to-High Extrapolation** We further probe the boundary of transferability by comparing against the previous strongest meta-training baseline under two adversarial regimes: **sparse training coverage** (Fig. 20, Challenge 2) and **reversed optimal regions** between training and testing (Fig. 21, Challenge 3). Across both stress tests, our method degrades far more gracefully than NAP and recovers quickly, showing that text-based reasoning learns a deeper extrapolation strategy that remains robust under adversarial low-fidelity regimes. Full setup, figures, and analysis are in Appendix C.6.

**Broader Impact and Future** This work contributes to broader AI Scientist (Lu et al., 2024). Our framework can be extended to more LLM configuration scenarios and broadly to any domain where cheap trials can guide expensive decisions (catalyst optimization, etc (Tom et al., 2024; Gupta et al., 2025)). Future directions: 1) expanding our Gym with more tasks and fidelity; 2) multi-objective optimization that balances competing goals. More broadly, **toward Recursive LLM Design,** as foreshadowed in our introduction, AutoLLMResearch is a concrete step toward Recursive Self-Improvement (Good, 1966; Schmidhuber, 2006; Zhuge et al., 2026; Rank et al., 2026): an LLM-based agent that accumulates experience from cheap experiments and uses it to inform expensive ones, effectively letting LLMs participate in the design of larger LLMs. Although a small step within a broader long-horizon agenda, we view this kind of cumulative experiential learning as a critical enabler of LLM recursive self-improvement, and a practical lever for accelerating and assisting human researchers as model and experiment costs continue to scale.

## 7 Conclusion

We tackle the **promising yet challenging** problem of automating high-cost LLM experiment configuration under strict budget constraints, which prior methods, all designed for low-cost trial-and-error settings, have never addressed. To our knowledge, we are **the first to propose an agentic training framework**, **AutoLLMResearch**, that learns generalizable principles from low-fidelity experiments and extrapolates them to expensive high-fidelity settings, supported by *LLMConfig-Gym* and a *structured training pipeline*. Extensive experiments confirm its effectiveness, generalization, and interpretability, offering a practical path toward scalable real-world LLM experiment automation.

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

## Appendix

## Appendix Contents

## A   LLMConfig-Gym: Tasks and Dataset Construction

LLMConfig-Gym combines open-source experiment datasets with ∼4,000 GPU hours of in-house GRPO tuning runs across multiple model sizes and datasets, yielding a unified offline environment spanning four representative LLM configuration tasks.

### A.1   LLMConfig-Gym Overview

### A.2   Unified Interface of LLMConfig-Gym

LLMConfig-Gym is implemented as a lookup-table-based environment for fast, deterministic evaluation of configuration policies (Fig. 7). All four tasks are exposed through a unified API; a simplified code snippet is shown in Fig. 8. The core functions are:

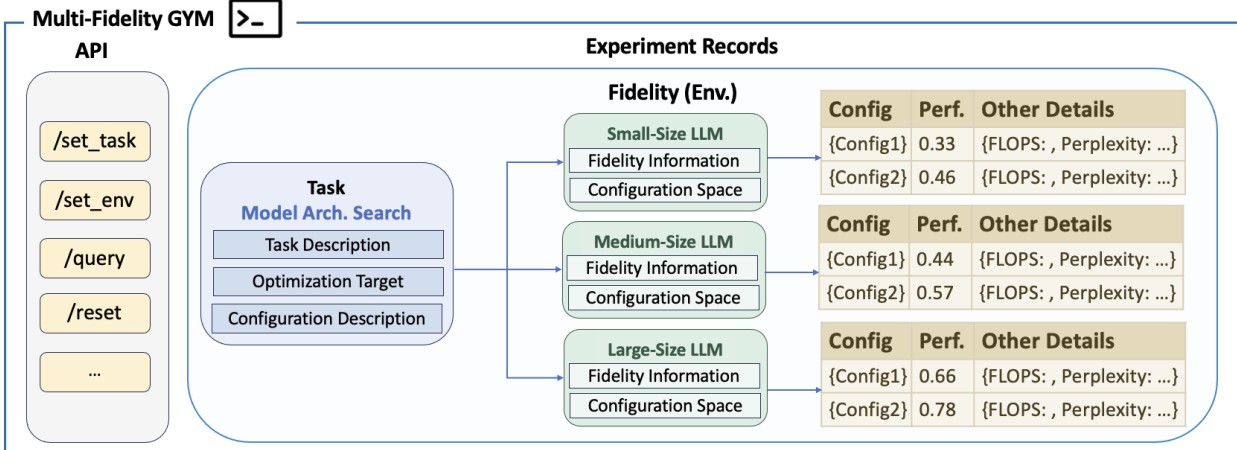

Figure 7: **LLMConfig-Gym overview.** A unified, lookup-table-based Gym organized by **Task → Fidelity → Experiment**.

- `list_tasks`: List all available configuration tasks (architecture search, data mixture, pretraining hyperparameters, RL hyperparameters).

- `set_task`: Select the active configuration task.

- `show_envs`: List available environments for the current task.

- `set_env`: Fix a concrete environment (e.g., dataset, model, training size).

- `show_configuration_space`: Return the configuration space for the current task and environment.

- `query`: Given a configuration (by ID or dictionary), return the target metric (e.g., perplexity, loss, aggregated score) and additional textual information (training score array, etc.) from the lookup table.

- `init_datasets`: Initialize the predefined training/testing datasets; users can also define custom datasets by setting different environments.

### A.3 Task 1: Model Architecture Configuration

Building on the 10k architecture evaluations and MLP perplexity surrogate from HW-GPT-Bench (Sukthanker et al., 2024) across the GPT-S/M/L family, we construct a model architecture configuration task in LLMConfig-Gym. The agent chooses GPT-2-style models parameterized by model scale (GPT-S/M/L), embedding dimension ($e$), number of layers ($l$), per-layer attention heads $\{h_t\}$, per-layer MLP ratios $\{m_t\}$, and a global bias flag, following the HW-GPT-Bench configuration space. For each configuration $s$, the Gym returns its validation perplexity on OpenWebText2 (Gao et al., 2020) from HW-GPT-Bench logs or the MLP surrogate, which downstream RL or black-box optimizers can turn into their own reward or acquisition functions. Task details are in Table 5.

### A.4 Task 2: Pretraining Hyperparameter Configuration

Building on the Step Law pre-training study (Li et al., 2025), which trains over 3,700 LLMs from scratch across seven model sizes $N$ and five dataset sizes $D$, we construct a pre-training hyperparameter tuning task in LLMConfig-Gym. The tunable hyperparameters are the peak learning rate LR and the global token batch size BS, while $(N, D)$ act as environment variables controlling the fidelity of each run. For each fixed $(N, D)$, the Gym exposes Step Law's grid: LR from a logarithmic sequence of powers of two, $\mathrm{LR} \in \{2^{-10.5}, 2^{-10.0}, \ldots, 2^{-7.0}\}$, and BS from a geometric progression in $\{32{,}768, \ldots, 4{,}194{,}304\}$ tokens/step

```
 1  from lmconfig_gym import LMConfigGym
 2
 3  # 1. Initialize and select RL tuning task
 4  env = LMConfigGym()
 5  print("Available tasks:", env.list_tasks())
 6  # Available tasks: ['rl_grpo_tuning', 'model_architecture_configuration',
 7  #     'pre-training_hyperparameter_tuning', 'training_data_mixing_configuration']
 7  env.set_task("rl_grpo_tuning")  # GRPO hyperparameter tuning
 8
 9  # 2. Inspect and fix task environment: dataset + model
10  env_space = env.show_envs()
11  # env_space = {
12  #     "dataset": ["gsm8k", "dapo", "mmlu_chemistry", "mmlu_history", "mmlu_physics"],
13  #     "model":   ["qwen2.5-1.5B-Instruct", "qwen2.5-3B-Instruct"],
14  #     "training_size": [256, 768, 1536]
15  #     "epoch": [15, 30]
16  # }
17  env.set_env(dataset="gsm8k", model="qwen2.5-3B-Instruct", training_size=256, epoch=30)
18
19  # 3. Inspect configuration space (hyperparameter grid)
20  config_space = env.show_configuration_space()
21  # config_space = {
22  #     "lr":         [1e-6, 5e-6, 1e-5],  # learning rate
23  #     "batch_size": [16, 32, 64],        # GRPO batch size
24  #     "kl_coef":    [0.0, 1e-3],         # KL regularization coefficient
25  # }
26
27  # 4. Query a single configuration
28  cfg = {"lr": 5e-5, "batch_size": 32, "kl_coef": 1e-3}
29  res = env.query(config=cfg)
30  # res = {
31  #     "score": 0.73,  # target metric (e.g., aggregated accuracy)
32  #     "additional_information": {
33  #         "critic_score_per_step": [...],  # per-step critic score
34  #         "...": "optional textual summary of training info",
35  #     },
36  # }
```

Figure 8: **Example usage of the `LMConfigGym` API for RL GRPO Tuning Configuration (Task 3).** The lookup-based interface supports task discovery, environment selection (dataset/model/training_-size/epoch), configuration-space inspection, and querying configurations to obtain both target metrics and additional information.

(ratio $\sqrt{2}$). For each configuration, the Gym returns the final smooth training loss, validated by Step Law as an unbiased proxy for validation loss. This enables downstream RL or black-box optimization methods to learn hyperparameter policies that adapt across model and data scales. Task details are in Table 6.

## A.5 Task 3: RL GRPO Tuning Configuration

To support agent training on LLM RL tuning, a widely deployed but expensive workflow, we collect an offline dataset of GRPO runs at 15 and 30 epochs across widely used datasets (GSM8K (Cobbe et al., 2021), DAPO-Math-17k (Yu et al., 2025), MMLU-Pro (Wang et al., 2024)) and two backbones (Qwen2.5-1.5B-Instruct and Qwen2.5-3B-Instruct), via grid search over critical GRPO hyperparameters to cover diverse combinations. All runs use 4 nodes $\times$ 4 NVIDIA A100 80G GPUs and total $\sim$4,000 GPU hours. Task details are in Table 7.

Table 5: **Task 1: Model Architecture Configuration in LLMConfig-Gym**, built on top of the HW-GPT-Bench dataset.

| Category | Item | Description |
|---|---|---|
| Environment (Fidelity) | Model Scale | One of `GPT-S`, `GPT-M`, `GPT-L`, as defined in HW-GPT-Bench (up to ∼1.55B parameters). |
| Configuration Space | Embedding dim. $e$ | Embedding dimension choices for each model, taken directly from HW-GPT-Bench; GPT-S: $e \in \{192, 384, 768\}$, GPT-M: $e \in \{256, 512, 1024\}$, GPT-L: $e \in \{320, 640, 1280\}$. |
| | Layers $l$ | Number of Transformer blocks from the depth set of the chosen model; GPT-S: $l \in \{10, 11, 12\}$, GPT-M: $l \in \{22, 23, 24\}$, GPT-L: $l \in \{34, 35, 36\}$. |
| | Per-layer heads $\{h_1, \ldots, h_l\}$ | For each layer, the number of attention heads is drawn from the allowed head set of that model; GPT-S: $h_t \in \{4, 8, 12\}$, GPT-M: $h_t \in \{8, 12, 16\}$, GPT-L: $h_t \in \{8, 16, 20\}$, for each layer $t$. |
| | Per-layer MLP ratios $\{m_1, \ldots, m_l\}$ | For each layer, the MLP ratio is chosen from the MLP-ratio set of the model; $m_t \in \{2, 3, 4\}$, for each layer $t$. |
| | Bias flag $b$ | $b \in \{\mathrm{On}, \mathrm{Off}\}$ for all models, indicating whether linear layers use bias. |
| Outputs | Target metric | For each architecture $s$, the Gym returns validation perplexity $\mathrm{PPL_{val}}(s)$ on OpenWebText2 and corresponding latency (s). |
| Meta-features | Task description | Large-model configuration optimization: the agent proposes architectures, observes $\mathrm{PPL_{val}}$ and Latency, and learns to select better configurations. |
| | Optimization target | Primary target is the normalized sum of validation perplexity $\mathrm{PPL_{val}}$ and Latency. The normalization is done by dividing the value by the maximum value of the target metric in the configuration space. |
| | Config. description | Structured and human-readable descriptions of each configuration (model scale, $e$, $l$, heads, MLP ratios, bias) for use as features or text prompts. |

To our knowledge, this is the first offline Gym for large-model RL configuration tuning; beyond the Gym, we also release the full Weights & Biases (W&B) logs to benefit the community.

## A.6 Task 4: Data Mixture Configuration

Building on ADMIRE IFT Runs (Ouyang et al., 2025), a dataset of 460 full instruction-finetuning runs on the Tülu 3 collection (Lambert et al., 2025) using Qwen2.5 at 500M, 3B, and 7B parameters across 256 data mixtures, we construct a data mixture configuration task in LLMConfig-Gym. The agent picks one of the 256 precomputed mixtures together with a model scale $m \in \{\mathtt{Qwen2.5\text{-}500M}, \mathtt{Qwen2.5\text{-}3B}, \mathtt{Qwen2.5\text{-}7B}\}$. For each configuration $\pi$, the Gym returns the average overall (ID + OOD) performance score across the 17 Tülu 3-style benchmarks. Task details are in Table 8.

## A.7 Task Split on LLMConfig-Gym in Our Paper

To probe cross-fidelity extrapolation, we define a **Low-Fidelity to High-Fidelity (L2H)** setting that orchestrates the collected Gym experiments: agents are trained on low-fidelity experiments and evaluated on high-fidelity ones. The detailed task splits are in Table 9.

Table 6: **Task 2: Pretraining Hyperparameter Configuration in LLMConfig-Gym**, built on top of the Step Law pre-training hyperparameter sweeps.

| Category | Item | Description |
|---|---|---|
| Environment (Fidelity) | Model and data scale | Model size $N$ (non-vocabulary parameters) and dataset size $D$ (number of training tokens). |
| Configuration Space | Learning rate LR | Peak learning rate selected from a logarithmic sequence of powers of two, with exponents from $-10.5$ to $-7.0$ in increments of 0.5, i.e., LR $\in \{2^{-10.5}, 2^{-10.0}, \dots, 2^{-7.0}\}$. |
| | Batch size BS | Global token batch size selected from a geometric progression, ranging from 32,768 to 4,194,304 tokens per step, with each subsequent value multiplied by $\sqrt{2}$ relative to the previous one. |
| Outputs | Target metric | For each configuration (LR, BS) given $N$, $D$, the Gym returns the final smooth training loss. |
| Meta-features | Task description | Pretraining Hyperparameter Configuration: the agent proposes (LR, BS) under given $(N, D)$, observes the resulting loss, and learns to select hyperparameters that generalize across model and data scales. |
| | Optimization target | Primary target is low smooth validation loss at high-fidelity settings (e.g., large $N$ and $D$), with low-fidelity runs providing cheaper proxy information. |
| | Config. description | Human-readable descriptions of each configuration (conditioning variables $N$ and $D$, and hyperparameters LR and BS) for use as structured features or as text prompts for LM-based agents. |

## A.8 Multi-Fidelity Optimization Landscape Analysis

We present the optimization landscape per task: Task 1 (Fig. 9), Task 2 (Fig. 10), Task 3 (Fig. 11), and Task 4 (Fig. 12). These landscapes also visualize the two cross-fidelity shifts identified in the introduction: **Tasks 1 and 4 reflect Challenge 2 (configuration space shift)**, where the training and testing landscapes lie over disjoint configuration grids; **Tasks 2 and 3 reflect Challenge 3 (optimization landscape shift)**, where the configuration space is (partially) shared across fidelities but the optimal region visibly moves between the training and testing heatmaps.

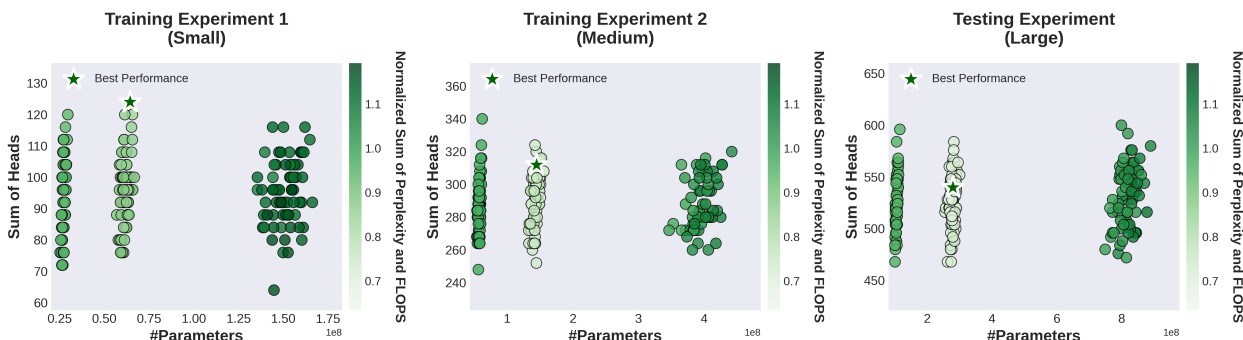

Figure 9: **Optimization landscape example for Task 1: Model Architecture Configuration in LLMConfig-Gym.** Each point corresponds to a sampled model configuration, colored by normalized performance (e.g., validation perplexity or regret). The rugged structure illustrates the presence of multiple local minima and the complexity of the search space.

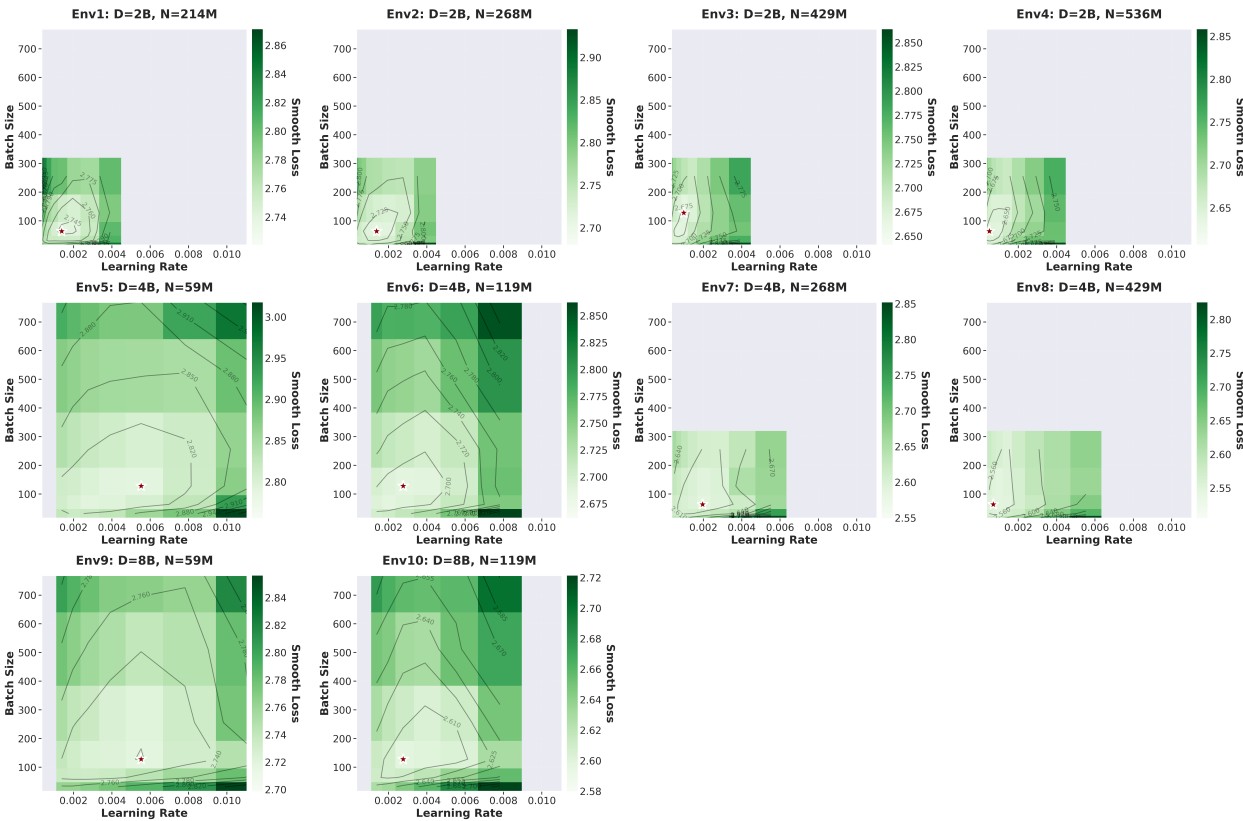

(a) Pre-training hyperparameter tuning, training set. Each point is a sampled (LR, BS) under fixed $(N, D)$, colored by smooth training loss; the heatmap highlights regions of optimal performance.

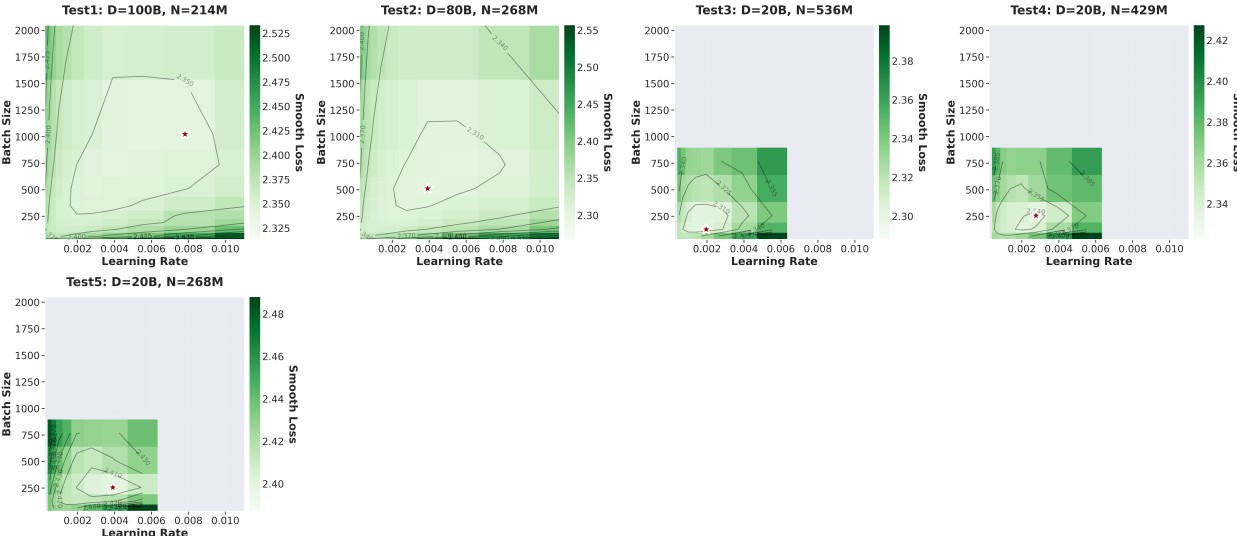

(b) Pre-training hyperparameter tuning, test set. Each point is a sampled (LR, BS) under fixed $(N, D)$, colored by smooth training loss; the heatmap highlights regions of optimal performance.

Figure 10: **Optimization landscapes for Task 2: Pretraining Hyperparameter Configuration in LLMConfig-Gym.** (a) shows the landscape on the training set, and (b) on the test set.

Table 7: **Task 3: RL GRPO Tuning Configuration in LLMConfig-Gym**, built from offline GRPO runs on Qwen2.5-1.5B-Instruct and Qwen2.5-3B-Instruct over GSM8K, DAPO-Math-17k, and MMLU-Pro with different training sizes and epochs.

| Category | Item | Description |
|---|---|---|
| Environment (Fidelity) | Model size and Training Data size | Model size: `Qwen2.5-1.5B-Instruct` or `Qwen2.5-3B-Instruct`; Dataset: `GSM8K`, `DAPO-Math`, and `MMLU-Pro`; Sampled Training size: $\{256, 768, 1536\}$, Training Epoch: $\{15, 30\}$. |
| Configuration Space | Learning rate LR | GRPO learning rate; LR $\in \{1 \times 10^{-6}, 5 \times 10^{-6}, 1 \times 10^{-5}\}$. |
| | Batch size BS | RL batch size (per update); BS $\in \{16, 32, 64\}$. |
| | KL regularization $\lambda_{\mathrm{KL}}$ | Coefficient on the KL-penalty term between policy and reference model; $\lambda_{\mathrm{KL}} \in \{0,\ 1 \times 10^{-3}\}$. |
| Outputs | Target metric | For each hyperparameter configuration $s = (\mathrm{LR}, \mathrm{BS}, \lambda_{\mathrm{KL}})$ and fidelity level, the Gym returns the aggregated evaluation score (e.g., average accuracy) on the test set, used as the optimization signal. |
| Meta-features | Task description | GRPO-based RL hyperparameter tuning: the agent proposes $s = (\mathrm{LR}, \mathrm{BS}, \lambda_{\mathrm{KL}})$, observes task-level and aggregated performance, and learns to select hyperparameters for large-model RL training. |
| | Optimization target | Primary target is high aggregated evaluation performance at the high-fidelity (Qwen2.5-3B-Instruct) setting. |
| | Config. description | Human-readable descriptions of each configuration (chosen LR, BS, $\lambda_{\mathrm{KL}}$, model scale, and dataset subset) for use as structured features or text prompts in LM-based agents. |

Table 8: **Task 4: Data Mixture Configuration in LLMConfig-Gym**, built on top of the ADMIRE IFT Runs dataset for Tülu 3 and Qwen2.5 models.

| Category | Item | Description |
|---|---|---|
| Environment (Fidelity) | Model size | Model size: `Qwen2.5-500M`, `Qwen2.5-3B`, or `Qwen2.5-7B`. |
| Configuration Space | Data mixture $\pi$ | Mixture over the Tülu 3 instruction datasets; in our offline Gym, $\pi$ is chosen from the 256 precomputed mixtures in ADMIRE IFT Runs. |
| Outputs | Target metric | For each configuration $s$, the Gym returns the average *overall* (ID + OOD) performance score across the 17 Tülu 3-style benchmarks, as reported in ADMIRE IFT Runs. |
| Meta-features | Task description | Instruction-finetuning data mixture configuration: Given the targeted instruct-tuned model, the agent proposes $\pi$, observes overall (ID+OOD) performance, and learns to select mixtures to maximize overall performance. |
| | Optimization target | Average overall (ID + OOD) performance score. |
| | Config. description | Human-readable descriptions of each configuration (mixture index or weights over named Tülu 3 datasets, and model scale) for use as structured features or text prompts. |

Table 9: **Low-Fidelity to High-Fidelity (L2H) Task Splits in our experiments using LLMConfig-Gym.**

| Task | Training Experiments (Low-Fidelity) | Testing Experiments (High-Fidelity) |
|---|---|---|
| Task 1: Model Architecture | GPT-M | GPT-L |
| Task 2: Pretraining Hyperparameter ($D$: Training Tokens $N$: Model Parameters) | D is 2000000000, N is 268304384
D is 2000000000, N is 429260800
D is 2000000000, N is 536872960
D is 4000000000, N is 59968512
D is 4000000000, N is 119992320
D is 4000000000, N is 268304384
D is 4000000000, N is 429260800
D is 8000000000, N is 59968512
D is 8000000000, N is 119992320 | D is 100000000000, N is 214663680
D is 80000000000, N is 268304384
D is 20000000000, N is 536872960
D is 20000000000, N is 429260800
D is 20000000000, N is 268304384 |
| Task 3: RL GRPO Tuning | MMLU_Chemistry(256 training samples), Qwen2.5-3B, Epoch:15
MMLU_History(256 training samples), Qwen2.5-3B, Epoch:15
MMLU_Physics(256 training samples), Qwen2.5-3B, Epoch:15
MMLU_Math(256 training samples), Qwen2.5-3B, Epoch:30 | GSM8K(768 training samples), Qwen2.5-3B, Epoch:30
GSM8K(1536 training samples), Qwen2.5-3B, Epoch:30
DAPO(768 training samples), Qwen2.5-3B, Epoch:30
DAPO(1536 training samples), Qwen2.5-3B, Epoch:30 |
| Task 4: Data Mixture | Qwen2.5-3B | Qwen2.5-7B |

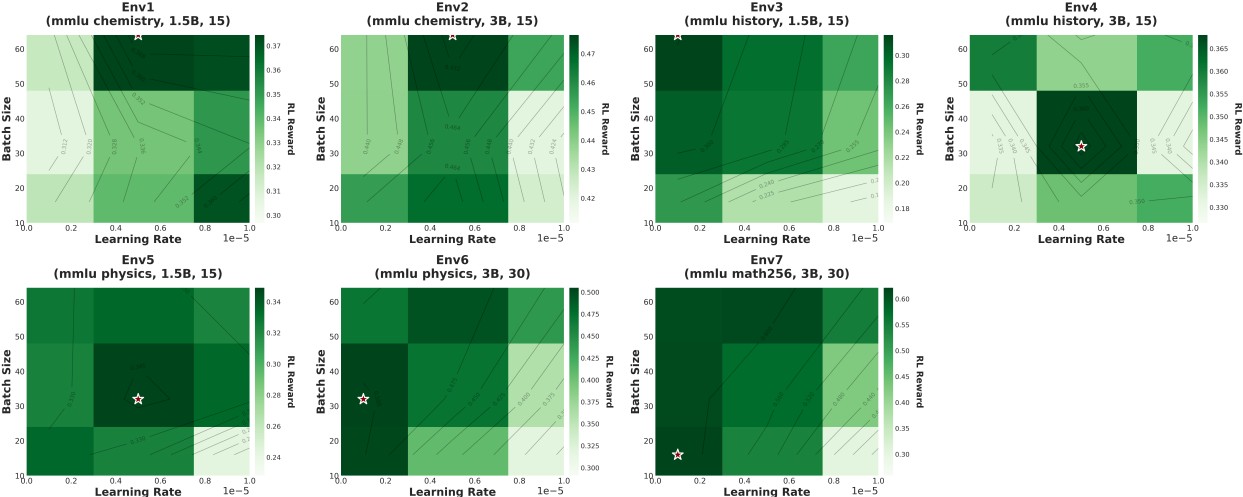

(a) Optimization landscape for GRPO RL-tuning (training set). Each point visualizes a unique $(LR, BS, \lambda_{KL})$ configuration for a fixed model and dataset, colored by the evaluated accuracy.

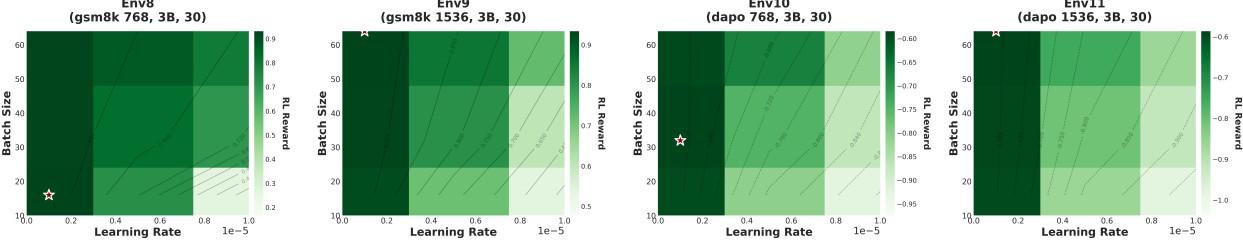

(b) Optimization landscape for GRPO RL-tuning (test set). Each point visualizes a unique $(LR, BS, \lambda_{KL})$ configuration for a fixed model and dataset, colored by the evaluated accuracy.

Figure 11: **Optimization landscapes for Task 3: RL GRPO Tuning Configuration in LLMConfig-Gym.** (a) shows the landscape on the training set, and (b) on the test set. These visualizations illustrate the challenge of hyperparameter tuning in RL and the structure of the reward surface.

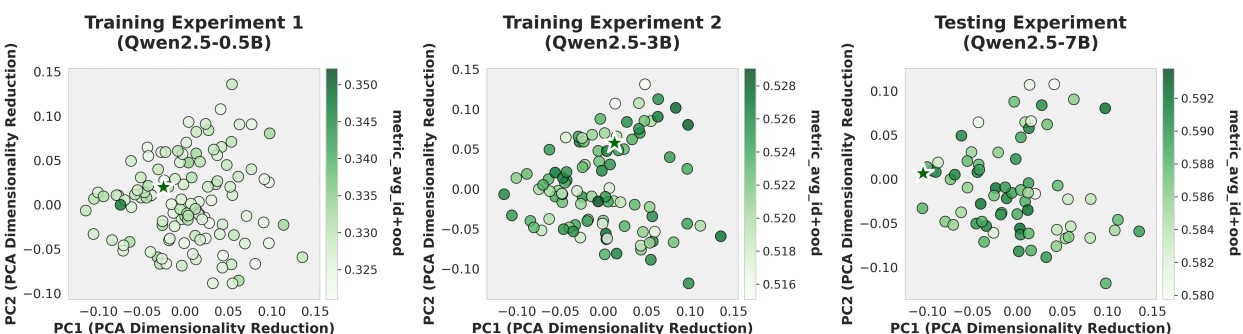

Figure 12: **Optimization landscape for Task 4: Data Mixture Configuration in LLMConfig-Gym.** Each point is a unique mixture configuration (proportions over data sources), colored by evaluation score, illustrating the effect of mixture ratios and the complexity of the search.

# B Implementation Details

## B.1 Prompts and Meta-Features

The prompts (Instructions and Context Meta-Features) we used for different tasks are as follows:

---

### Prompt for Task 1: Model Architecture Configuration

**You are a Large Language Model architecture expert. Your task is to optimize a Transformer model architecture to balance (1) training perplexity on the dataset and (2) FLOPs.**
**Optimization target**: Perplexity bounds: - min (best) perplexity = 18.02 - max (worst) perplexity = 30.33. FLOPs bounds: - min FLOPs = 245883994112 - max FLOPs = 2584367595520 . We minimize the following objective (min–max normalized sum): Target = (perplexity - 18.02) / (30.33 - 18.02) + (flops - 245883994112) / (2584367595520 - 245883994112), Lower Target is better.
**Configuration space (must strictly follow)** The configuration space for the Transformer model architecture is as follows: Here are the meanings of the configuration space: - sample_embed_dim: embedding dimension of the Transformer model - sample_n_layer: number of layers of the Transformer model - sample_n_head: number of attention heads in each layer of the Transformer model (per-layer list) - sample_mlp_ratio: ratio of MLP dimension to embedding dimension of the Transformer model (per-layer list) - sample_bias: whether to use bias in the MLP of the Transformer model ("True"/"False" as a string)
**Here are the candidate model architectures (numbered options)**:
Option 1: 'sample_embed_dim': 320, 'sample_n_layer': 35, 'sample_n_head': [8, 16, 8, 16, 8, 20, 16, 8, 8, 8, 16, 16, 16, 16, 16, 20, 20, 16, 16, 8, 8, 8, 8, 20, 8, 8, 16, 20, 20, 8, 20, 16, 8, 16, 20, 8], 'sample_mlp_ratio': [4, 4, 2, 2, 2, 4, 4, 2, 4, 3, 3, 2, 2, 2, 2, 4, 2, 4, 4, 4, 3, 3, 4, 4, 4, 2, 2, 2, 4, 3, 3, 3, 3, 4, 2],  'sample_bias': 'True'
........
Option 20: 'sample_embed_dim': 1280, 'sample_n_layer': 35, 'sample_n_head': [16, 20, 20, 8, 20, 20, 16, 8, 8, 8, 16, 20, 8, 16, 8, 20, 8, 20, 8, 20, 8, 16, 8, 8, 16, 20, 20, 16, 20, 20, 20, 16, 20, 8, 16, 8], 'sample_mlp_ratio': [2, 2, 2, 2, 2, 3, 2, 2, 3, 3, 4, 4, 2, 4, 4, 2, 4, 4, 2, 4, 4, 3, 2, 3, 4, 2, 2, 4, 2, 4, 3, 3, 4, 3, 4, 4], 'sample_bias': 'False'
**Output requirements (CRITICAL - violations will cause errors)** 1) Select EXACTLY ONE architecture from the provided candidate architectures above. Do NOT create any new values and do NOT edit any candidate. 2) Output ONLY the final configuration dict (no extra text, no other words), wrapped between <config> and </config> tags.
**Now Please think step by step, select the best architecture dict that achieves the lowest Target, and return the configuration dict between <config> and </config> tags.** Think in 4 Steps: Step 1: Propose (Generate 5 Candidates) Based on domain knowledge, previous results, and remaining budget, generate 5 diverse, promising configurations exploring different regions of the search space. Note the configuration must be in the configuration space of the task! Step 2: Imagine (Value Estimation) For each candidate, estimate its performance by reasoning about: - Previous experimental results including the previous proposed configurations and their scores - Training dynamics (convergence, stability) - Model capacity and generalization - Hyperparameter interactions The score should be estimated score based on the previous experimental results, training dynamics, model capacity, and hyperparameter interactions. Step 3: Select Best Select the most promising configuration based on value estimations, domain knowledge, and previous results. Explain your choice and output it between <config> and </config> tags. Step 4: Call "exec_config" tool to get the score of the configuration in the final when you obtain config between <config> and </config> tags. You MUST have to call "exec_config" tool to query the score of the configuration in the final when you obtain config between <config> and </config> tags!
**From previous low-fidelity experiments, here are previous related environment and Top5 score configurations which may be helpful for you to propose the next promising configuration.** ###### Experiment Environment information: Medium Transformer architecture search, configuration space: "sample_embed_dim": one from [256, 512, 1024], "sample_n_layer": one from [22, 23, 24], "sample_n_head": for each layer, one from [8, 12, 16], "sample_mlp_ratio": for each layer, one from [2, 3, 4], "sample_bias": one from ["True", "False"], Top-5 configurations: 1. 'sample_embed_dim': 512, 'sample_n_layer': 23, 'sample_n_head': [16, 12, 12, 12, 16, 12, 16, 12, 12, 16, 8, 16, 12, 16, 12, 16, 12, 16, 8, 16, 8, 8, 16, 12], 'sample_mlp_ratio': [4, 2, 2, 2, 3, 4, 3, 2, 3, 4, 3, 3, 4, 3, 4, 3], 'sample_bias': 'True' ...... 5. 'sample_embed_dim': 512, 'sample_n_layer': 24, 'sample_n_head': [8, 8, 12, 16, 8, 16, 8, 16, 8, 16, 16, 16, 12, 8, 12, 16, 12, 16, 16, 16, 12, 12, 12, 16], 'sample_mlp_ratio': [4, 3, 2, 4, 4, 2, 3, 3, 4, 3, 3, 3, 3, 4, 3, 4, 4, 2, 4, 4, 4, 3, 4, 4], 'sample_bias': 'True' ######
**Remember**: 1. Consider your remaining budge is 1 , previous experimental results, best configurations from low-fidelity experiments when making decisions. 2. You MUST have to call "exec_config" tool to query the score of the configuration in the final when you obtain config between <config> and </config> tags!

---

### Prompt for Task 2: Pretraining Hyperparameter Configuration

**You are a Large language model instruction tuning expert, your task now is to optimize the instruction tuning.** The model is trained on the dataset the total number of training tokens seen by the model during training is: 100000000000 and the count of trainable model parameters excluding token embedding matrices is: 214663680. These two numbers influence the best learning rate and batch size of the instruction tuning.
**Optimization target** The optimization target is the smooth loss of the model training on dataset. The lower the smooth loss, the better.
**Configuration space (must strictly follow)** The configuration space for the instruction tuning is: "'lr': [0.007812, 0.0004883, 0.0002441, 0.0006905, 0.001953, 0.0009766, 0.001381, 0.0003453, 0.003906, 0.01105, 0.005524, 0.002762], 'bs': [736, 2048, 32, 128, 1024, 256, 512, 352, 192, 64]" In this configuration space: - lr: learning rate - bs: batch size
**Output requirements (hard constraints)** 1) All values must strictly follow the configuration space. 2) Output ONLY the final configuration dict (no extra text, no other words), wrapped between <config> and </config> tags.
**Now Please think step by step and propose the best configuration dict that achieves the lowest Target, and return the configuration dict between <config> and </config> tags.** Think in 4 Steps: Step 1: Propose (Generate 5 Candidates) Based on domain knowledge, previous results, and remaining budget, generate 5 diverse, promising configurations exploring different regions of the search space. Note the configuration must be in the configuration space of the task! Step 2: Imagine (Value Estimation) For each candidate, estimate its performance by reasoning about: - Previous experimental results including the previous proposed configurations and their scores - Training dynamics (convergence, stability) - Model capacity and

generalization - Hyperparameter interactions The score should be estimated score based on the previous experimental results, training dynamics, model capacity, and hyperparameter interactions. Step 3: Select Best Select the most promising configuration based on value estimations, domain knowledge, and previous results. Explain your choice and output it between <config> and </config> tags. Step 4: Call "exec_config" tool to get the score of the configuration You MUST have to call "exec_config" tool to query the score of the configuration in the final when you obtain config between <config> and </config> tags!

**From previous low-fidelity expperiments, here are previous related environment and Top3 score configurations which may be helpful for you to propose the next promising configuration.** Experiment Environment information: the total number of training tokens seen by the model during training is: 2000000000, and the count of trainable model parameters excluding token embedding matrices is: 214663680 In this environment, the Top-3 configurations are: 1. learning rate: 0.001381, batch size: 64.0 2. learning rate: 0.001953, batch size: 64.0 3. learning rate: 0.001953, batch size: 128.0 ###### ...... ###### Experiment Environment information: the total number of training tokens seen by the model during training is: 8000000000, and the count of trainable model parameters excluding token embedding matrices is: 119992320 In this environment, the Top-3 configurations are: 1. learning rate: 0.002762, batch size: 128.0 2. learning rate: 0.003906, batch size: 128.0 3. learning rate: 0.005524, batch size: 128.0 ######

**Remember**: 1. Consider your remaining budge is 2 , previous experimental results, best configurations from low-fidelity experiments when making decisions. 2. You MUST have to call "exec_config" tool to query the score of the configuration in the final when you obtain config between <config> and </config> tags!

## Prompt for Task 3: RL GRPO Tuning Configuration

**You are a Large language model RL-GRPO tuning expert, your task now is to optimize the RL-GRPO tuning.** The model is Reinforcement learning trained on gsm8k dataset with 1536 samples, the base model is Qwen2.5-3B-Instruct, and the training epoch is 30.

**Optimization target:** The optimization target is the validation score of the RL-GRPO model training. The higher the validation score, the better.

**Configuration space (must strictly follow)** The configuration space for the RL-GRPO tuning is: "'lr': [1e-06, 5e-06, 1e-05], 'mb': [16, 32, 64], 'kl': [0, 0.001]" In this configuration space: - lr: learning rate of the RL-GRPO training - mb: mini-batch size for the gradient update of the RL-GRPO training - kl: weight of the KL divergence of the RL-GRPO training

**Output requirements (hard constraints)** 1) All values must strictly follow the configuration space. 2) Output ONLY the final configuration dict (no extra text), wrapped between <config> and </config> tags.

**Now Please think step by step and propose the best configuration dict that achieves the highest validation score, and return the configuration dict between <config> and </config> tags.** Think in 4 Steps: Step 1: Propose (Generate 5 Candidates) Based on domain knowledge, previous results, and remaining budget, generate 5 diverse, promising configurations exploring different regions of the search space. Note the configuration must be in the configuration space of the task! Step 2: Imagine (Value Estimation) For each candidate, estimate its performance by reasoning about: - Previous experimental results including the previous proposed configurations and their scores - Training dynamics (convergence, stability) - Model capacity and generalization - Hyperparameter interactions The score should be estimated score based on the previous experimental results, training dynamics, model capacity, and hyperparameter interactions. Step 3: Select Best Select the most promising configuration based on value estimations, domain knowledge, and previous results. Explain your choice and output it between <config> and </config> tags. Step 4: Call "exec_config" tool to get the score of the configuration You MUST have to call "exec_config" tool to query the score of the configuration in the final when you obtain config between <config> and </config> tags!

**From previous low-fidelity expperiments, here are previous related environment and Top3 score configurations which may be helpful for you to propose the next promising configuration.** Experiment Environment information: the dataset is mmlu_chemistry, the model is Qwen2.5-1.5B-Instruct, Note the Training epoch is 15 In this environment, the Top-3 configurations are: 1. kl loss weight: 0.001, learning rate: 1e-05, batch size: 64.0 2. kl loss weight: 0.001, learning rate: 5e-06, batch size: 64.0 3. kl loss weight: 0.0, learning rate: 5e-06, batch size: 32.0 ###### ...... ###### Experiment Environment information: the dataset is mmlu_history, the model is Qwen2.5-1.5B-Instruct, Note the Training epoch is 15 In this environment, the Top-3 configurations are: 1. kl loss weight: 0.0, learning rate: 1e-06, batch size: 64.0 2. kl loss weight: 0.0, learning rate: 1e-06, batch size: 32.0 3. kl loss weight: 0.001, learning rate: 5e-06, batch size: 64.0 ######

**Remember**: 1. Consider your remaining budge is 1 , previous experimental results, best configurations from low-fidelity experiments when making decisions. 2. You MUST have to call "exec_config" tool to query the score of the configuration in the final when you obtain config between <config> and </config> tags!

## Prompt for Task 4: Data Mixture Configuration

**You are an expert in data-centric optimization for large language model training.** Your task is to optimize the training data mixture for a target language model by maximizing the metric_avg_id+ood. The model is trained on the dataset with different data mixture ratios. The base model is Qwen2.5-7B.

**Problem formulation** We consider $d$ training data domains. A data mixture is a vector $\boldsymbol{\pi} = (\pi_1, \ldots, \pi_d)$ lying on the probability simplex: $\pi_i \geq 0$, $\sum_{i=1}^{d} \pi_i = 1$. The metric_avg_id+ood of a data mixture $\boldsymbol{\pi}$ is defined as the average of performance metrics in id (internal distribution) and ood (out-of-distribution) data domains:

$$\text{metric\_avg\_id+ood}(\boldsymbol{\pi}) = \frac{1}{2} \left( \text{metric\_avg\_id}(\boldsymbol{\pi}) + \text{metric\_avg\_ood}(\boldsymbol{\pi}) \right)$$

where metric_avg_id($\boldsymbol{\pi}$) is the average performance metric of the model trained on the id data domains, and metric_avg_ood($\boldsymbol{\pi}$) is the average performance metric of the model trained on the ood data domains.

The goal is to find the data mixture $\boldsymbol{\pi}$ that maximizes metric_avg_id+ood($\boldsymbol{\pi}$). Note that metric_avg_id+ood($\boldsymbol{\pi}$) is the average of performance metrics in id and ood data domains. The higher the metric_avg_id+ood($\boldsymbol{\pi}$), the better.

**Configuration space (must strictly follow)** The configuration space for the data mixture tuning is an array of ratios of different data domains. You must choose π from a discrete candidate set. The columns names (this defines the order of the array entries) for each ratio are: "['ratio_coconot_converted', 'ratio_evol_codealpaca_heval_decontaminated', 'ratio_flan_v2_converted', 'ratio_no_robots_converted', 'ratio_numinamath_tir_math_decontaminated', 'ratio_oasst1_converted', 'ratio_per-

sonahub_code_v2_34999', 'ratio_personahub_ifdata_manual_seed_v3_29980', 'ratio_personahub_math_v5_regen_149960', 'ratio_tulu_hard_coded_repeated_10', 'ratio_tulu_v3.9_aya_100k', 'ratio_tulu_v3.9_open_math_2_gsm8k_50k', 'ratio_tulu_v3.9_personahub_math_interm_algebra_20k', 'ratio_tulu_v3.9_sciriff_10k', 'ratio_tulu_v3.9_synthetic_finalresp_wildguardmixtrain_decontaminated_50k', 'ratio_tulu_v3.9_table_gpt_5k', 'ratio_tulu_v3.9_wildchat_100k', 'ratio_tulu_v3.9_wildjailbreak_decontaminated_50k', 'ratio_tulu-3-sft-personas-math-grade']"

**Here are the candidate data mixtures (numbered options), each mixture is an array of floats in [0, 1] corresponding to the above domain order**: Option 1: [0.0108, 0.0976, 0.0344, 0.0462, 0.0498, 0.0062, 0.045, 0.0112, 0.0942, 0.0, 0.209, 0.072, 0.0172, 0.0248, 0.0176, 0.008, 0.0534, 0.0614, 0.1412] ...... Option 16: [0.0212, 0.0464, 0.0378, 0.0062, 0.023, 0.0052, 0.0232, 0.0476, 0.2244, 0.0002, 0.098, 0.0164, 0.0884, 0.0134, 0.1002, 0.0076, 0.0944, 0.0482, 0.0982]

**Output requirements (hard constraints)** 1) Select EXACTLY ONE mixture from the provided candidate mixtures above. Do NOT create any new values and do NOT edit any candidate. 2) Output ONLY the final configuration array (no extra text), wrapped between <config> and </config> tags.

**Now Please think step by step, select the best data mixture from the provided candidates that maximizes metric_avg_id+ood, and return the configuration array between <config> and </config> tags.** Think in 4 Steps: Step 1: Propose (Generate 5 Candidates) Based on domain knowledge, previous results, and remaining budget, generate 5 diverse, promising configurations exploring different regions of the search space. Note the configuration must be in the configuration space of the task! Step 2: Imagine (Value Estimation) For each candidate, estimate its performance by reasoning about: - Previous experimental results including the previous proposed configurations and their scores - Training dynamics (convergence, stability) - Model capacity and generalization - Hyperparameter interactions The score should be estimated score based on the previous experimental results, training dynamics, model capacity, and hyperparameter interactions. Step 3: Select Best Select the most promising configuration based on value estimations, domain knowledge, and previous results. Explain your choice and output it between <config> and </config> tags. Step 4: Call "exec_config" tool to get the score of the configuration You MUST have to call "exec_config" tool to query the score of the configuration in the final when you obtain config between <config> and </config> tags!

**From previous low-fidelity expperiments, here are previous related environment and Top5 score configurations which may be helpful for you to propose the next promising configuration.** Experiment Environment information: The data mixture ratios on Qwen2.5-3B, the size of the model is 3B Top-5 configurations: 1. [0.004, 0.085, 0.0568, 0.006, 0.0268, 0.0086, 0.052, 0.053, 0.1746, 0.0008, 0.1336, 0.095, 0.0116, 0.0258, 0.0466, 0.0078, 0.1366, 0.0586, 0.0168] ......

**Remember**: 1. Consider your remaining budge is 5 , previous experimental results, best configurations from low-fidelity experiments when making decisions. 2. You MUST have to call "exec_config" tool to query the score of the configuration in the final when you obtain config between <config> and </config> tags!

## B.2 Interactions between Agent and LLMConfig-Gym as Function Calling

The exact tool-call schema (used by the verl function calling mechanism in our main-text setup) is shown below.

**Tool Configuration**

```
tools:
- class_name: "verl.tools.configcoder_tool.ConfigcoderTool"
  config: {}
  tool_schema:
    type: "function"
    function:
      name: "exec_config"
      description: "A tool for executing configuration dict or array and return the execution results"
      parameters:
        type: "object"
        properties:
          config:
            type: "string"
            description: "The ONLY generated configuration dict or array between <config> and </config> tags that
will be executed"
        required: ["config"]
```

## B.3 Hyperparameter Settings For Training

A summary of the training setup is provided in the main text. The full hyperparameter dumps for Policy Distillation and End-to-End GRPO training are shown in the following code snippets.

**Hyperparameters for Policy Distillation (SFT)**

```
finetuning_type: full
template: qwen3
cutoff_len: 23000
per_device_train_batch_size: 1
gradient_accumulation_steps: 4
learning_rate: 5.0e-6
```

```
num_train_epochs: 8.0
lr_scheduler_type: cosine
warmup_ratio: 0.1
bf16: true
```

## Hyperparameters for RL

```
--config-path="$CONFIG_PATH" \
--config-name='configcoder_multiturn_grpo' \
custom_reward_function.path=$REWARD_PATH  \
algorithm.adv_estimator=grpo \
data.train_files=$DATA_FOLDER/train${data_suffix}.parquet \
data.val_files=$DATA_FOLDER/test${data_suffix}.parquet \
data.train_batch_size=32 \
data.max_prompt_length=8500 \
data.max_response_length=13000 \
data.filter_overlong_prompts=True \
data.truncation='error' \
data.return_raw_chat=True \
actor_rollout_ref.model.path=$MODEL_PATH \
actor_rollout_ref.actor.optim.lr=1e-6 \
actor_rollout_ref.model.use_remove_padding=True \
actor_rollout_ref.actor.ppo_mini_batch_size=16 \
actor_rollout_ref.actor.ppo_micro_batch_size_per_gpu=4 \
actor_rollout_ref.actor.use_kl_loss=True \
actor_rollout_ref.actor.kl_loss_coef=0.001 \
actor_rollout_ref.actor.kl_loss_type=low_var_kl \
actor_rollout_ref.actor.entropy_coeff=0 \
actor_rollout_ref.model.use_fused_kernels=False \
actor_rollout_ref.actor.use_dynamic_bsz=True \
actor_rollout_ref.actor.ppo_max_token_len_per_gpu=30000 \
actor_rollout_ref.rollout.log_prob_use_dynamic_bsz=true \
actor_rollout_ref.rollout.log_prob_max_token_len_per_gpu=34000 \
actor_rollout_ref.ref.log_prob_use_dynamic_bsz=true   \
actor_rollout_ref.ref.log_prob_max_token_len_per_gpu=34000 \
actor_rollout_ref.model.enable_gradient_checkpointing=True \
actor_rollout_ref.actor.fsdp_config.param_offload=False \
actor_rollout_ref.actor.fsdp_config.optimizer_offload=False \
actor_rollout_ref.rollout.log_prob_micro_batch_size_per_gpu=32 \
actor_rollout_ref.rollout.tensor_model_parallel_size=1 \
actor_rollout_ref.rollout.name=sglang \
actor_rollout_ref.rollout.gpu_memory_utilization=0.8 \
actor_rollout_ref.rollout.n=8 \
actor_rollout_ref.ref.log_prob_micro_batch_size_per_gpu=32 \
actor_rollout_ref.ref.fsdp_config.param_offload=True \
algorithm.use_kl_in_reward=False \
trainer.critic_warmup=0 \
trainer.logger=['console','wandb'] \
trainer.project_name='verl_grpo_configcoder' \
trainer.experiment_name="${data}${data_suffix}_${concrete_model}" \
trainer.val_before_train=True \
trainer.n_gpus_per_node=4 \
trainer.nnodes=1 \
trainer.save_freq=5 \
trainer.test_freq=5 \
trainer.validation_data_dir="./${data}${data_suffix}_${concrete_model}_rollouts_configcoder_train/" \
actor_rollout_ref.rollout.multi_turn.tool_config_path="$PROJECT_DIR/examples/sglang_multiturn/config/tool_config/
configcoder_tool_config.yaml" \
trainer.total_epochs=${TOTAL_EPOCHS}$
```

# C  Additional Experimental Results

## C.1  RQ1: Performance for Different Tasks

We report per-task average regret across budgets 1–5 for all baselines on the four LLMConfig-Gym tasks (Figs. 13, 14, 15, and 16).

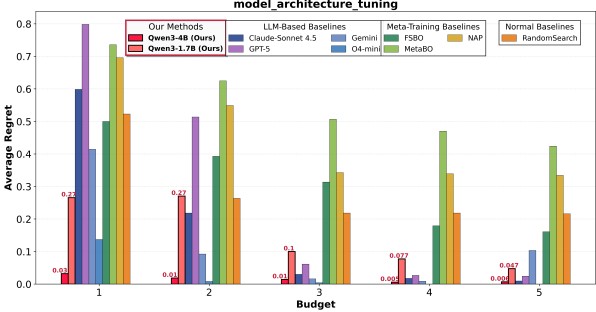

Figure 13: **Performance on Task 1: Model Architecture Configuration.**

Figure 14: **Performance on Task 2: Pretraining Hyperparameter Configuration.**

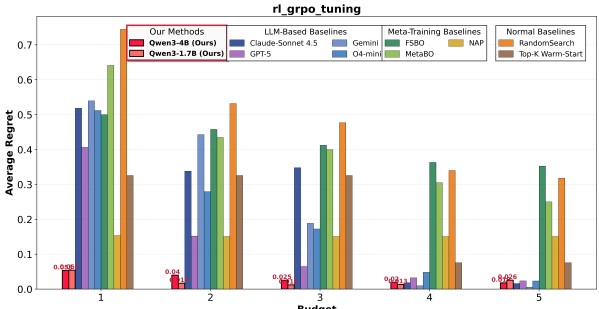

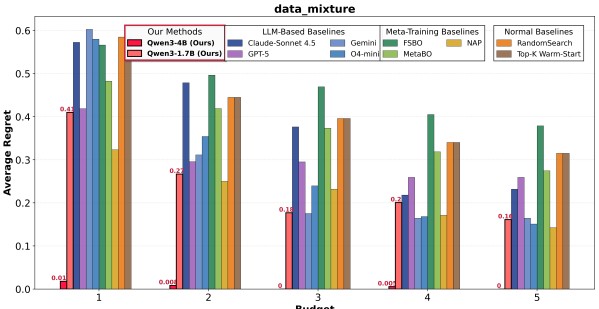

Figure 15: **Performance on Task 3: RL GRPO Tuning Configuration.**

Figure 16: **Performance on Task 4: Data Mixture Configuration.**

## C.2  RQ2-1: Full Case Study Details (Configuration Space Shift and Optimization Landscape Shift)

We provide here the full reasoning-trajectory analysis for both case studies summarized in the main text.

**Case Study 1: Configuration Space Shift (Model Architecture).** The training and test configuration spaces differ substantially (e.g., `embed_dim` shifts from $\{256, 512, 1024\}$ to $\{320, 640, 1280\}$; `n_layer` from $\{22–24\}$ to $\{34–36\}$), so configurations optimal on the training set do not even exist on the test set. **(1) Before Training.** The base agent reasons heuristically on both sets: it flags FLOPs concerns for large `embed_dim` but fails to analyze the perplexity vs. FLOPs trade-off, defaulting to the minimum (`embed_dim`=256 on training, 320 on test). The reasoning is essentially circular and yields poor scores on both (0.7313 train / 0.9464 test). **(2) During Training.** On the training set, RL rewards drive the agent toward explicit multi-objective analysis: "*lower `embed_dim` reduces FLOPs but may hurt perplexity; higher values help perplexity but increase FLOPs*". The agent then selects the moderate `embed_dim`=512 with a diverse architecture. Crucially, the reward shapes a *principle* ("balance extremes"), not a fixed configuration. **(3) Inference on Test Set.** Applying the same principle to the shifted test space, the agent selects the moderate `embed_dim`=640 (the middle value in $\{320, 640, 1280\}$), even though this exact configuration never appeared during training. The test score improves from 0.9464 to 0.6627 (a 30% relative gain), showing that what transfers across the configuration-space shift is not a configuration but a *scaling-aware balancing rule*.

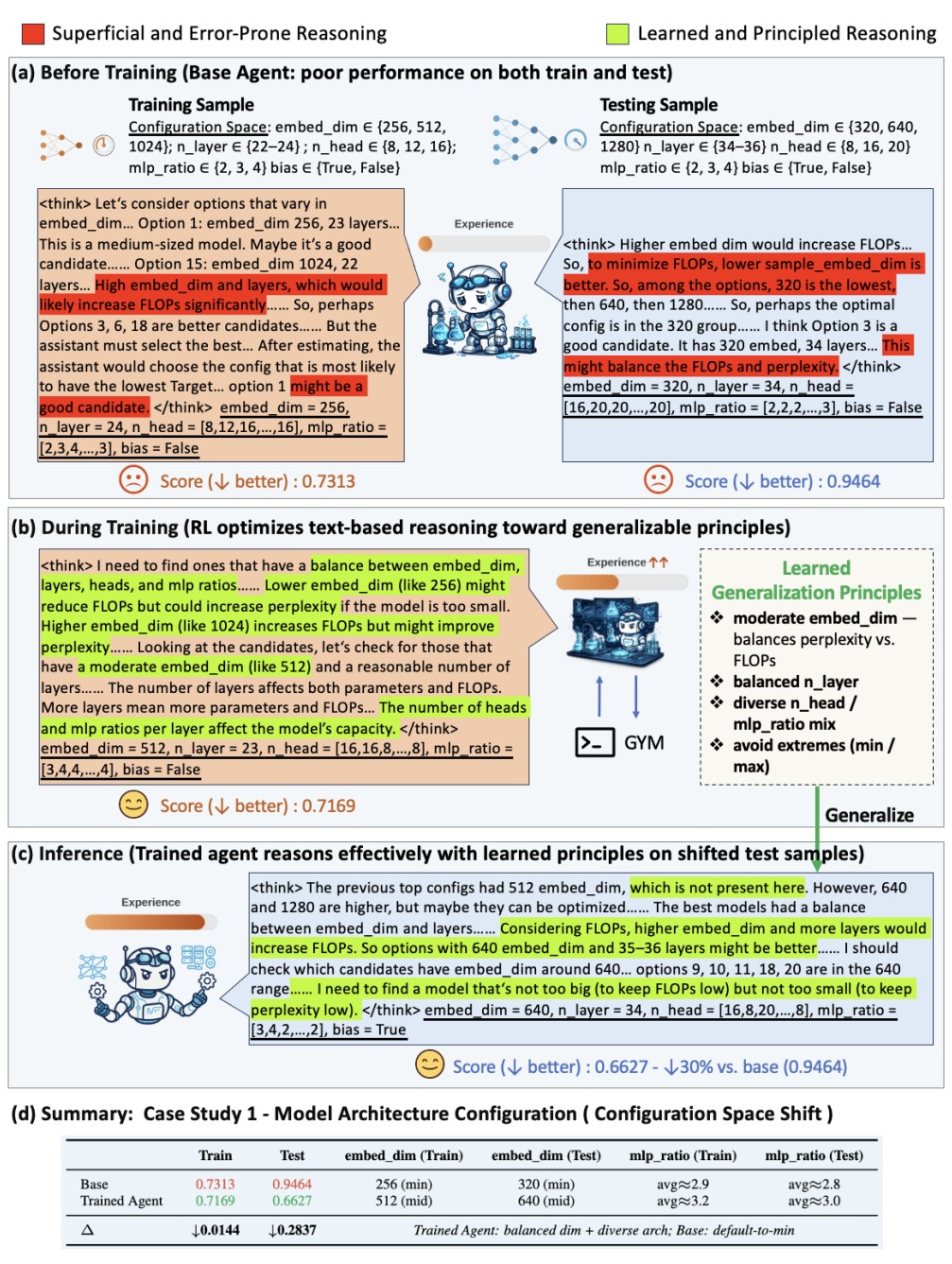

Figure 17: **Case Study 1 on Model Architecture Configuration, targeting Challenge 2 (Configuration Space Shift).** Reasoning trajectories of the base agent (*Before Training*) and our RL-trained agent (*During Training* and *Inference*) on the training set (embed_dim ∈ {256, 512, 1024}, n_layer ∈ {22–24}) and the disjoint test set (embed_dim ∈ {320, 640, 1280}, n_layer ∈ {34–36}). Highlighted phrases mark key reasoning steps. The base agent defaults to the minimum embed_dim on both sets, whereas the RL-trained agent learns a balanced-architecture principle (moderate embed_dim, balanced n_layer, diverse n_head/mlp_ratio mix) that transfers to the shifted test space, yielding ∼30% lower regret on the test set.

**Case Study 2: Optimization Landscape Shift (Pretraining Hyperparameter).** Training and testing share the same configuration variables (`lr`, `bs`) but differ in fidelity (2B vs. 20B tokens), causing the optimization landscape to shift. **(1) Before Training.** On the training set, the base agent grounds its choice in vague claims about "previous top configs" without analyzing actual prior experiments. At test time it is worse: it latches onto an oversimplified trend ("more tokens ⇒ higher lr"), entirely ignores the parameter–lr relationship, and selects the highest available `lr`=5.5e-3 for a 536M-parameter model, the wrong direction. **(2) During Training.** On the training set, the agent learns to reason jointly over fidelity variables: "*higher parameters lead to lower lr to avoid divergence; more tokens allow slightly higher lr*". Rather than memorizing a single best configuration, it internalizes *fidelity-dependent trends* (e.g., at 268M params the best lr was 1.95e-3; at 429M, 6.9e-4) and uses them as a reasoning scaffold. **(3) Inference on Test Set.** At test time with 536M params and 20B tokens, the agent recalls the learned trend, interpolates across fidelities ("for 536M params the lr should be between 1.95e-3 and 6.9e-4, adjusted slightly upward for more tokens"), and selects `lr`=1.95e-3, `bs`=128. This yields a 0.67% gain over the base agent while avoiding its catastrophic over-extrapolation, showing that what transfers across the optimization-landscape shift is a *fidelity-dependent scaling trend.*

### C.3   RQ2-2: From Optimization Perspective: Text-Based Reasoning Prunes Search Space

Beyond qualitative reasoning, we examine whether the agent's improved reasoning translates into measurable *optimization effectiveness.* To isolate the contributions of our two design modules (Experiment Curation and RL Learning with the Gym), we compare three variants on the same Pretraining Hyperparameter Configuration task (Task 2): **Qwen3-4B-Base (w/o experiment curation)**, **Qwen3-4B-Base**, and **Qwen3-4B (Ours, trained agent)**. For each method, we plot the trajectory R1→R2→R3 ($R_k$ is the $k$-th round). Since each run is repeated three times, we further draw a dashed ellipse enclosing all explored configurations across repeated runs, characterizing the *effective search region.* The background heatmap is the ground-truth loss across the full configuration space. Three distinct behaviors emerge in Fig. 6: **(1) Without experiment curation**, the agent has no low-fidelity context and explores erratically, jumping from `lr=7.5e-4, bs=256` to `lr=4e-3, bs=768` without analyzing R1's feedback. Its search region is the largest, covering optimal and suboptimal configurations indiscriminately. This indicates that without fidelity-aware curation, the agent lacks the context to make informed decisions and falls back to random-like exploration; the wide, unfocused region visually confirms that the budget is wasted across the entire space rather than directed at promising areas. **(2) The base model without training** reads low-fidelity trends but extrapolates in the *wrong direction*: it over-generalizes "*lower lr is better for more parameters*" and progressively drifts toward overly conservative `lr=5e-4, bs=64` in R2, moving away from the optimum. Low-fidelity context alone is insufficient: without RL training to calibrate the reasoning process, the agent latches onto surface heuristics that lead to systematic errors. Its search region is narrower than variant (1) but is concentrated in the wrong area, showing that incorrect extrapolation can be worse than no extrapolation at all. **(3) Our trained agent** correctly extrapolates from low-fidelity experience: it lands at `lr=1.5e-3, bs=128` in R1 (already near the global best), then makes an informed incremental adjustment to `lr=2e-3, bs=128` in R2 to reach the global optimum (loss 2.302). Its search region is tightly concentrated around the optimum, confirming that RL training on text-based reasoning yields an agent that both identifies a strong starting point from the very first iteration and refines it through principled increments rather than broad exploration or surface heuristics.

Together, these results show that text-based reasoning learned from experience acts as a *search prior*: it provides a strong initial configuration from the first round and enables correct trend extrapolation that prunes the search space (ours), whereas incorrect extrapolation or lack of grounding yields poor initializations and wastes budget in suboptimal regions (other variants). This complements the reasoning-level case studies in Appendix C.2, confirming that the transferable principles learned by our agent translate directly into measurable search efficiency gains. The three variants jointly validate our two-module design: Experiment Curation provides the necessary fidelity-aware context, and RL training with the Gym calibrates the agent's reasoning to leverage that context for effective extrapolation.

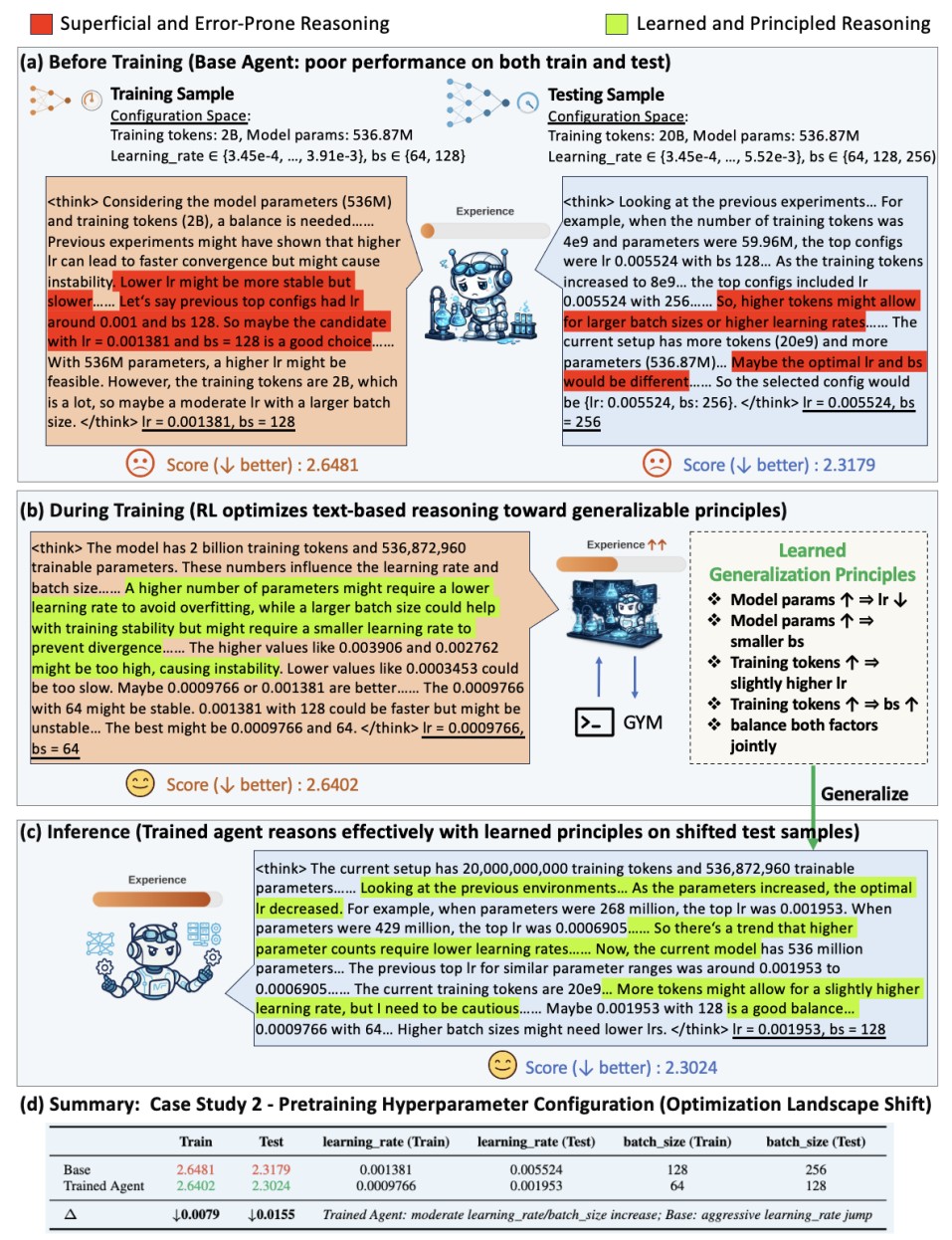

Figure 18: **Case Study 2 on Pretraining Hyperparameter Configuration, targeting Challenge 3 (Optimization Landscape Shift).** Reasoning trajectories on the training set (low-fidelity, 2B tokens) and the test set (high-fidelity, 20B tokens) under the same configuration variables ($\mathtt{lr}, \mathtt{bs}$) but different fidelities. The base agent ignores the parameter–lr scaling trend and over-extrapolates lr at test time, while the RL-trained agent learns fidelity-dependent trends (params $\uparrow \Rightarrow$ lr $\downarrow$; tokens $\uparrow \Rightarrow$ slightly higher lr; tokens $\uparrow \Rightarrow$ bs can $\uparrow$) and produces a calibrated configuration that improves over the base agent.

## C.4   Amortized Cost Analysis: Estimation Methodology

For each task, we measure the GPU hours of our training experiments (low-/medium-fidelity grid search) and testing experiments (high-fidelity), recording the performance our method achieves at budget = 5. We then compute the total GPU hours each baseline (LLM-based prompting, Optuna) requires to match this performance from scratch on every new task. Our per-task cost is *Training GPU Hours + Average Testing GPU Hours*; baselines pay only *Average Testing GPU Hours*, but repeatedly on every new task. We thus compute baselines' cumulative cost as the product of the average per-task GPU hours (measured on our collected tasks) and the number of tasks, producing a linear curve by construction. For our method, we add a one-time upfront training cost on top of the same average per-task testing cost, reflecting its negligible marginal cost per additional task. Since our test tasks are limited in number, we extrapolate to 30 tasks using the average per-task testing cost. This analysis illustrates the scalability advantage of our method, not precise savings for any specific deployment; in practice, the upfront meta-learning cost can be absorbed by **idle GPU time**, since the low-/medium-fidelity training experiments are offline and can be scheduled whenever spare compute is available, and these grid-search experiments are ones researchers would typically conduct to explore scaling laws and tuning principles.

## C.5   RQ3: Training Dynamics — Figure, Metric Definitions, and Full Analysis

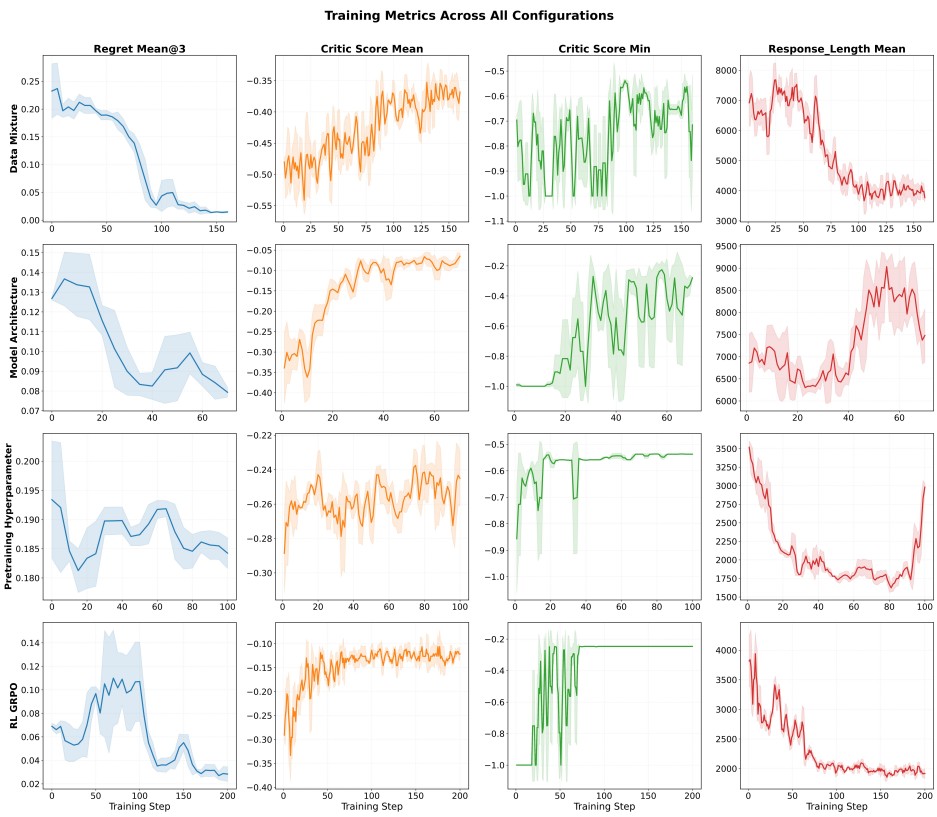

Figure 19: **Training dynamics across all four tasks**, showing **Regret Mean@3**, **Critic Score Mean**, **Critic Score Min**, and **Response Length Mean** over training steps.

We list here the precise definitions of the four metrics we partition training samples by task type and track throughout training:

- **Regret Mean@3**: average regret on the held-out test set, measuring whether learned principles extrapolate to unseen high-fidelity experiments.

- **Critic Score Mean**: average reward on training samples, reflecting configuration quality on tasks the agent is trained on.

- **Critic Score Min**: minimum reward across training samples per batch, capturing worst-case behavior and the rate of invalid configurations.

- **Response Length Mean**: average response length (tokens), indicating whether the agent adapts its reasoning complexity to task types.

Fig. 19 shows the following trends across all four task categories. **Regret Mean@3 decreases consistently**, indicating that the agent progressively proposes higher-quality configurations; since regret is measured on the held-out high-fidelity test set, this confirms that principles learned from low-/medium-fidelity training experiments successfully extrapolate. **Critic Score Mean rises steadily** (gradually rather than in sudden jumps), suggesting the agent is internalizing transferable principles rather than memorizing specific configurations. **Critic Score Min** is particularly informative: for tasks with complex configuration spaces (Model Architecture, Task 1; RL GRPO Tuning, Task 3), it initially stays at $-1.0$ due to frequent format violations but gradually rises, confirming the agent learns to avoid catastrophic failures. **Response Length Mean** varies by task: longer trajectories for tasks requiring nuanced trade-off analysis (e.g., Model Architecture's multi-dimensional search space), shorter outputs for simpler spaces, a task-adaptive behavior that emerges naturally from RL training without explicit length supervision.

## C.6 Stress-Testing the Boundary of Low-to-High Extrapolation

We further probe the boundary of transferability by comparing against the previous strongest meta-training baseline under two adversarial regimes: **sparse training coverage** (Fig. 20, Challenge 2) and **reversed optimal regions** between training and testing (Fig. 21, Challenge 3).

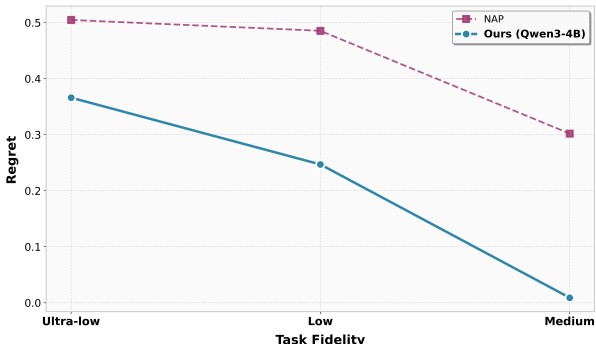

Figure 20: **Sparse configuration space coverage (Challenge 2)** on the **Data Mixture Configuration** task (Task 4). Training experiments are reduced to "Ultra-low" ($\sim$20%), "Low" ($\sim$40%), and full "Medium", with the testing set fixed. Our method consistently achieves lower regret than NAP and declines far more steeply as coverage grows.

Figure 21: **Reversed optimal region (Challenge 3)** on the **RL GRPO Tuning Configuration** task (Task 3). "Ultra-low" contains only experiments whose optimum is the *worst* at test time; "Low" mixes these with non-adversarial ones; "Medium" matches the main setup. Our method degrades gracefully and reaches near-zero regret ($\sim$0.02) at "Medium" vs. NAP's $\sim$0.08.

**Sparse Configuration Space Coverage (Challenge 2).** We stress-test the agent's ability to learn generalizable principles across configuration spaces (Challenge 2) by progressively reducing the number of training experiments, making the covered configuration distribution increasingly misaligned with the testing space. We design this on the Data Mixture Configuration task (Task 4): "Ultra-low" fidelity contains only $\sim$20% of the original training experiments, "Low" contains $\sim$40%, and "Medium" matches our main setup; the testing set is held fixed across all three levels. Fig. 20 shows that (1) both learning methods (ours and NAP) decrease regret as training fidelity increases, but ours declines far more steeply; (2) even at "Ultra-low"

fidelity, our method still outperforms NAP, which struggles to learn from such sparse data. This indicates that NAP rigidly memorizes training configurations and degrades in data-scarce regimes, whereas our method provides a substantially more robust mechanism for cross-fidelity extrapolation.

**Reversed Optimal Region (Challenge 3).** We further stress-test the agent's ability to capture configuration trends across fidelities (Challenge 3) by selecting training experiments whose optimal region is *reversed* relative to the testing set. We design this on the RL GRPO Tuning Configuration task (Task 3): "Ultra-low" fidelity contains only experiments whose optimum is the *worst* configuration at test time (e.g., learning rate $= 10^{-5}$); "Low" mixes these adversarial experiments with additional low-fidelity ones whose optima are closer to the reversed region; "Medium" matches our main setup. Fig. 21 shows that (1) NAP's regret at "Ultra-low" fidelity ($\sim 0.43$) is dramatically worse, confirming that NAP rigidly memorizes training configurations and is severely misled when the training optimum contradicts the testing one; (2) our method degrades at "Ultra-low" but recovers quickly at "Low" once non-adversarial experiments are mixed in and reaches near-zero regret ($\sim 0.02$) at "Medium" compared to NAP's $\sim 0.08$, outperforming NAP across all fidelity levels. These results demonstrate that our RL-based method does not memorize surface-level configuration–performance mappings; it learns a deeper extrapolation strategy that remains robust even when training and testing optimal regions diverge, a property critical for real-world deployment. **Result 10:** Our method degrades far more gracefully across both stress tests and recovers quickly, showing that text-based reasoning learns a deeper extrapolation strategy that remains robust under adversarial low-fidelity regimes.

# D  Theoretical Analysis of the Cost-Effective Trade-Off

The previous section showed empirical cost-effectiveness on the four LLMConfig-Gym tasks, reporting estimated GPU hours as the number of high-fidelity tasks grows. We now present a more general theoretical analysis, demonstrating the broader applicability of our framework and characterizing the conditions under which meta-training yields a net computational benefit.

**Notation.**  We define the following quantities to formalize the analysis:

- $K$: the number of target (test) tasks to be solved after deployment,

- $M$: the number of source tasks used during offline meta-training,

- $E_m$: the number of low-fidelity exploration steps consumed per source task during meta-training,

- $S_{\text{base}}$: the number of high-fidelity evaluations required for a baseline method (without meta-knowledge) to reach a target performance on a single task,

- $S_{\text{meta}}$: the number of high-fidelity evaluations required for the meta-trained model to reach the same target performance via few-shot adaptation,

- $\alpha = t_{\text{HF}}/t_{\text{LF}}$: the cost ratio between a single high-fidelity and a single low-fidelity evaluation.

For convenience, we denote the per-task saving in high-fidelity steps as $\Delta S = S_{\text{base}} - S_{\text{meta}}$.

**Break-even condition.**  Under the baseline approach, each of the $K$ target tasks must be optimized from scratch, yielding a total cost of

$$C_{\text{base}} = K \cdot S_{\text{base}} \cdot t_{\text{HF}}. \tag{2}$$

Our meta-learning framework incurs a one-time offline cost for exploring $M$ source tasks in the low-fidelity environment, followed by a per-task online adaptation cost:

$$C_{\text{meta}} = M \cdot E_m \cdot t_{\text{LF}} + K \cdot S_{\text{meta}} \cdot t_{\text{HF}}. \tag{3}$$

The meta-learning approach is cost-effective whenever $C_{\text{meta}} < C_{\text{base}}$, which simplifies to

$$M \cdot E_m < K \cdot \Delta S \cdot \alpha. \tag{4}$$

Inequality equation 4 makes explicit the interplay of three independent factors: (i) the *scale of deployment $K$*, which amplifies every unit of high-fidelity saving across future tasks; (ii) the *algorithmic gain $\Delta S$*, the reduction in high-fidelity samples the meta-model requires versus training from scratch; and (iii) the *environment cost ratio $\alpha$*, which converts each saved high-fidelity step into a large number of "free" low-fidelity steps.

**Task leverage ratio.**  Rearranging Inequality equation 4 by grouping the task-count variables on one side yields

$$\frac{K}{M} > \frac{E_m}{\Delta S \cdot \alpha}. \tag{5}$$

We refer to the left-hand side $K/M$ in Inequality equation 5 as the *task leverage ratio*: it captures how many target tasks each source task effectively subsidizes. The right-hand side is a *critical amortization threshold* determined entirely by algorithmic efficiency ($E_m$, $\Delta S$) and the physical cost structure ($\alpha$).

Two practical insights follow. First, when high-fidelity experiments are very expensive ($\alpha \gg 1$), even a modest $\Delta S$ pushes the right-hand-side threshold very low, so the meta-training investment pays off even when $K$ is comparable to or smaller than $M$. Second, when meta-training exploration is efficient (small $E_m$, e.g., via RL-guided search rather than exhaustive grid search), the amortization threshold drops further, broadening the regime in which our framework is advantageous.

**Critical task threshold.** For deployment planning, it is useful to solve Inequality equation 4 for the minimum number of target tasks required to break even:

$$K > \frac{M \cdot E_m}{\Delta S \cdot \alpha}.$$

(6)

Any deployment with $K$ exceeding the threshold in Inequality equation 6 guarantees a net reduction in total computational cost. In our experiments (Section 5), instantiating this formula with the measured $S_{\text{base}}$, $S_{\text{meta}}$, $E_m$, and $\alpha$ shows the break-even point is reached after only a small number of target tasks, confirming the practical cost-effectiveness of our approach.

