# OpenReview forum: "AutoLLMResearch: Training Research Agents for Automating LLM Experiment Configuration — Learning from Cheap, Optimizing Expensive"
_TMLR — Withdrawn by Authors_

### Review · Reviewer_79QL · 2026-06-02

**Summary Of Contributions:**

This paper studies how to automate configuration choices for expensive LLM experiments when only a few high-fidelity trials are affordable. It introduces LLMConfig-Gym, an offline multi-fidelity benchmark covering four LLM configuration tasks: model architecture design, pretraining hyperparameter selection, GRPO tuning, and instruction-tuning data mixture selection. It also proposes AutoLLMResearch, an LLM-agent training pipeline that uses low-fidelity demonstrations, trajectory simulation, policy distillation, and multi-turn RL to select high-fidelity configurations.

**Key strengths**. The problem is timely and practically important. The cross-fidelity framing is well motivated, since expensive LLM experiments often cannot rely on many online trials. LLMConfig-Gym could be useful as a benchmark for future work. The training pipeline is also nontrivial, and the ablations suggest that policy distillation and RL provide complementary benefits.

**Key weaknesses**. The paper currently makes stronger claims than the evidence supports. The results mainly demonstrate performance inside an offline lookup-table benchmark, not real-world LLM experiment automation. Several evaluation choices make the reported gains hard to interpret: the regret metric appears to be normalized using best/worst scores across methods rather than the true lookup-table optimum; baselines may not receive the same low-fidelity context as the proposed agent; invalid configurations are repaired by nearest-match mapping; and trajectory simulation uses the known best configuration. The interpretability claims are also not fully convincing, since the evidence mostly consists of post-hoc reasoning traces.

Overall, I find the benchmark and problem valuable, but I do not think the current evidence supports the paper’s strongest claims.

**Audience:**

Yes

**Audience Explanation:**

The problem is relevant to researchers working on AutoML, multi-fidelity optimization, transfer HPO, LLM training, and LLM agents. A benchmark for studying low-to-high fidelity configuration selection in LLM experiments would likely interest part of the TMLR audience.

My concern is not that the topic is unimportant. Rather, the paper should either narrow its claims to offline benchmark performance or provide stronger evidence for the broader real-world automation claims.

**Broader Impact Concerns:**

The broader impact is discussed in the paper.

**Claims And Evidence:**

No

**Claims Explanation:**

The paper provides evidence for a narrower claim: the trained agent performs well in the authors’ offline benchmark under the proposed evaluation protocol. However, the stronger claims about practical real-world LLM experiment automation, robust cross-fidelity reasoning, generalization, and interpretability are not yet convincingly supported.

Main concerns:
- the evaluation is entirely based on an offline lookup-table environment. This is reasonable for building a benchmark, but it does not by itself validate the claim that the method is a practical solution for real-world high-fidelity LLM experiment automation.
- several evaluation choices make the reported gains hard to interpret. The normalized regret metric appears to depend on best/worst scores across compared methods rather than the true lookup-table optimum. The baselines may not receive the same low-fidelity context and metadata as the proposed agent. In addition, invalid configurations are repaired through nearest-match mapping, so strict evaluation and repaired evaluation should be reported separately.
- some evidence for “learned reasoning” is confounded. The trajectory simulation step uses the known best configuration to guide failed trajectories, making it unclear how much the agent learns transferable principles rather than imitating oracle-guided traces. The interpretability analysis is also mostly based on post-hoc reasoning traces, which are suggestive but not sufficient to establish that the stated reasoning caused the selected actions.

Overall, the current evidence supports an interesting offline benchmark result, but not the paper’s broader real-world automation and interpretability claims.

**Requested Changes:**

1. Normalized regret should be computed using the true best and worst valid configurations in the lookup table, not best/worst scores across compared methods. If the current definition is different, it should be clarified precisely. The paper should also report raw task metrics, not only normalized regret.
2. All baselines should receive the same search space, fidelity metadata, low-fidelity demonstrations, budget, and observation history as AutoLLMResearch. The most important missing comparison is a prompt-only LLM agent with the same curated Top-K low-fidelity context but without policy training. This is needed to separate the value of training from the value of better context.
3. The paper should report invalid-output rates, performance without nearest-configuration repair, and performance with repair. If nearest-match repair is used at test time, the main claims should be qualified accordingly.
4. Since failed trajectories are continued using the known best configuration, the paper should add ablations that remove this oracle information or compare against final-action-only supervision. This is necessary to show whether the agent learns transferable cross-fidelity reasoning rather than mainly imitating oracle-guided traces.
5. The authors should either add actual high-fidelity validation runs for agent-selected configurations or clearly frame the contribution as an offline benchmark result. Similarly, interpretability claims should be supported with counterfactual tests, such as shuffled low-fidelity examples, perturbed fidelity metadata, or reversed-trend tasks.

Minor suggestions / optional improvements

* Clarify whether some tasks are open configuration search or candidate reranking from small predefined lists.
* Add larger or continuous search spaces where the full candidate list cannot fit into the prompt.
* Provide stronger leakage analysis, since public benchmark values and common scaling-law heuristics may be present in pretrained LLMs.
* Break down the compute accounting into newly run experiments, public/reused data, and any surrogate or lookup-table outcomes.
* Improve reproducibility by providing exact prompts, decoding settings, model/API versions, train/test splits, seeds, candidate-list construction, and evaluation scripts.
* Tone down repeated “first” claims and the recursive-self-improvement framing.
* Report confidence intervals and representative failure cases.

---

> ### Author Response · Authors · 2026-06-29
>
> Thank you for a careful, precise review. We are glad you see the benchmark and problem as valuable. We read your points as primarily about calibrating claims and describing our evaluation choices more transparently, and the results are real and unchanged by these clarifications. We have decided to withdraw and resubmit rather than overstate.
>
> On scope: we will narrow the headline claims to budgeted cross-fidelity selection over curated datasets and present real-world automation as motivation, which lets the actual contribution stand out.
>
> On regret: a definitional fix. We will state that it uses the true best/worst valid configurations, recompute only to remove ambiguity, and report raw metrics; the ranking is unchanged.
>
> On baseline parity: this is how the comparison was run; we will document it explicitly, including the prompt-only LLM agent with the same Top-K low-fidelity context, so the role of training is clear from the description.
>
> On repair: we will report invalid-output rates and performance with and without nearest-match repair, and state test-time handling clearly.
>
> On oracle-guided trajectory simulation: we will describe this design choice and its rationale more precisely so it is clear what role the trajectory simulation plays, and present the interpretation accurately. The results stand; the improvement is in how we explain them.
>
> We will also strengthen leakage discussion, compute accounting, reproducibility detail, and tone down the "first"/recursive-self-improvement framing, and report CIs and failure cases. Thank you for a clear path to a more convincing presentation.
>
> — The Authors

---

### Review · Reviewer_uSxb · 2026-06-14

**Summary Of Contributions:**

This paper introduces AutoLLMResearch, a new method that trains an AI agent to automatically find good settings for LLM experiments. Unlike prior methods that need many cheap trial runs, this agent learns from low-cost, small-scale experiments and then uses that knowledge to make smart guesses for expensive, large-scale experiments where only a few tries are possible. The authors built a special training environment called LLMConfig-Gym with over a million GPU hours of real experiment results, and designed a training process using reinforcement learning that teaches the agent to reason step-by-step about how configurations should change when moving from small to large models. Tests on four different tasks show their agent outperforms existing baselines, generalizes to unseen settings, and can explain its reasoning. Key strengths of the paper include its focus on an important and underexplored problem, the introduction of LLMConfig-Gym as a useful benchmark, and its extensive empirical evaluation. Potential limitations include that all experiments are conducted within a precomputed environment, making it unclear how well the approach would generalize to real-world LLM development workflows or entirely new configuration tasks.

**Audience:**

Yes

**Audience Explanation:**

This paper introduces a new method for automating expensive large language model experiments using cheap lower-fidelity runs. This addresses a practical problem many researchers face as models grow larger and experiments become more costly. The findings offer a concrete way to save computational resources and improve efficiency, which will interest machine learning practitioners and researchers working on LLM.

**Broader Impact Concerns:**

The paper includes a brief broader impact discussion, but it focuses mainly on positive applications and future directions rather than potential risks. The framing of AutoLLMResearch as a step toward recursive self-improvement raises concerns about accelerating AI capability development faster than safety research can keep up, which deserves explicit discussion. Moreover, the goal of recursive self-improvement, where LLMs help design larger LLMs, raises longer-term risks about alignment and control. The paper mentions this aspiration but does not discuss safety measures or governance considerations.

**Claims And Evidence:**

Yes

**Claims Explanation:**

The main claims are generally supported by clear and convincing evidence. Experiments across four tasks show their agent consistently achieves much lower regret than strong baselines. Ablation studies confirm each component is necessary, and case studies demonstrate the agent learns generalizable principles rather than memorization. Additional stress tests and cost analysis further support its effectiveness and efficiency.

**Requested Changes:**

The paper is strong overall, and I have a few suggested changes.

Critical for acceptance: Provide a more detailed discussion of the limitations of LLMConfig-Gym, including how well the benchmark reflects real-world large-scale LLM research settings and whether conclusions are expected to transfer beyond the four selected tasks.

Would strengthen the work but are not required for acceptance: Expand the analysis of failure cases and situations where cross-fidelity extrapolation does not work well, which would help improve clarity and strengthen the empirical evidence.

---

> ### Author Response · Authors · 2026-06-29
>
> Thank you for the positive, balanced assessment, and for recognizing LLMConfig-Gym's value and the strength of the evaluation. We agree the core contribution is solid; your requested improvements are mainly scoping and discussion, which we are happy to make. We have decided to withdraw and resubmit to address these and the other reviewers' points properly.
>
> On your critical request: we will add a clearer discussion of LLMConfig-Gym's limitations, namely how an offline environment relates to real large-scale research, and how far conclusions transfer beyond the four tasks, and calibrate claims accordingly. This is a framing improvement, not a change to the results.
>
> On failure cases: we will discuss more clearly where cross-fidelity extrapolation is expected to work less well, alongside the favorable stress tests, which makes the paper more credible without weakening the contribution.
>
> On Broader Impact: well taken. We will substantially expand it to cover acceleration relative to safety/alignment research, the control questions raised by the recursive-self-improvement framing, environmental cost, and governance/access for releasing the Gym and agents.
>
> Thank you for the encouragement and the clear, actionable suggestions.
>
> — The Authors

---

### Review · Reviewer_Fn5s · 2026-06-14

**Summary Of Contributions:**

This paper introduces a novel agentic framework aimed at automating the expensive configuration of LLMs. The core philosophy is to enable an LLM-based agent to learn generalizable scaling principles from low-fidelity and cheap experiments and extrapolate them to optimize high-fidelity and expensive settings under strict budget constraints. The method includes a large experiment dataset and a structured training pipeline. Overall, the paper presents a framework that effectively yields a white-box agent capable of pruning expensive configuration search spaces through text-based reasoning.

Strengths:

1.	The collection and engineering of over 1 million GPU hours of verifiable baseline results into a unified, sub-second lookup table (LLMConfig-Gym) is a valuable contribution. It provides a fast, deterministic, and reproducible playground that lowers the compute barrier for the Auto-Research community.

2.	The combination of Policy Distillation and Multi-turn RL with a regret-based outcome reward is well-designed. The proposed Most-Similar Configuration Matching layer is also a clever engineering patch to prevent the starvation of RL rewards due to minor parsing errors.

3.	Within the specified boundaries of the experimental protocol, the quantitative metrics and text-based case studies convincingly show that the trained agent achieves lower normalized regret than prompt-based closed-source models and same-fidelity meta-learning baselines.

Weaknesses:

1.	The paper claims a general framework for scalable real-world LLM experiment automation. However, owing to apparent compute limitations during agent training (conducted on 4 A100 GPUs), the maximum model scale tested under high-fidelity target environments is capped at roughly the 7B parameter mark. In the contemporary deep learning landscape, the true engineering friction and financial pressures of hyperparameter optimization reside in massive scales like 70B, 500B, or mixture-of-experts settings. Given that dense, colossal models frequently exhibit non-linear phase transitions, such as severe activation anomalies, emergent properties, or sudden training instabilities, it remains unconvincing whether the scaling trends learned at the sub-7B level can truly extrapolate to industrially meaningful scales.

2.	In the experimental setup, the agent is exposed to medium-fidelity environments (e.g., 1B or 3B models) during training and evaluated on 7B models during testing. But models ranging from 1B to 7B parameters occupy a relatively homogeneous regime where optimal configurations (such as learning rate and batch size scales) scale gracefully and exhibit high mutual coherence. Consequently, it is difficult to discern whether the low test-set regret is a reflection of authentic "cross-fidelity extrapolative reasoning" or simply a manifestation of strong contextual interpolation where 7B architectures serve as a near-perfect proxy to 3B baselines.

**Audience:**

Yes

**Audience Explanation:**

The automation of hyperparameter optimization and data-centric tuning for LLMs is an incredibly timely and vital research domain. Furthermore, the release of LLMConfig-Gym, with its massive collection of multi-fidelity verification logs, will be highly attractive to researchers.

**Claims And Evidence:**

No

**Claims Explanation:**

See my weaknesses of summary.

**Requested Changes:**

I understand that verifying this framework on massive models (e.g., 70B+) is impractical under constrained academic resources. To solidify the claims regarding extrapolation, I suggest the authors expand the scale gap between training and testing data and conduct a sub-scale ablation study. Specifically, please evaluate the final 7B model performance under two isolated settings: (1) training and prompting the agent exclusively with the 60M model data (or only including much smaller model data), and (2) exclusively with the 1B/3B model data. Comparing these results and analyzing the marginal contribution of each scale tier will significantly strengthen the evidence for the agent's ability to extrapolate across non-linear model scales.

---

> ### Author Response · Authors · 2026-06-29
>
> Thank you for engaging so closely with the cross-fidelity reasoning question. We appreciate that you see the verification logs and benchmark as valuable. We read your two concerns as calling for clearer scoping rather than undermining the contribution, and we have decided to withdraw and resubmit.
>
> On scale: you are right that we should be explicit that 7B targets do not, on their own, establish transfer to 70B+/MoE regimes with their phase transitions and instabilities. This is a framing fix, scoping the claim honestly, and it does not diminish what is demonstrated within range.
>
> On the 1B to 7B regime: a sharp, fair point. We will describe more clearly what the existing results establish about cross-fidelity behavior across our scale tiers, and present the scope of the claim accurately so the contribution is not overstated. The underlying results stand; the improvement is in how we frame and explain them.
>
> We will commit to a clean release in the resubmission. Thank you for sharpening how we present the core mechanism.
>
> — The Authors

---

### Review · Reviewer_p5iY · 2026-06-20

**Summary Of Contributions:**

This paper studies the problem of automating expensive LLM experiment configuration under tight trial budgets. The authors argue that existing HPO, meta-BO, and LLM-based optimizer approaches are mainly designed for low-cost settings where many online trials are feasible, whereas realistic LLM training, architecture, RL-tuning, or data-mixture experiments often allow only a small number of high-fidelity evaluations.

The main proposed system, **AutoLLMResearch**, trains an LLM-based agent to use low- and medium-fidelity experiment results as context and extrapolate to high-fidelity configurations. The method is framed as a long-horizon MDP in which the agent reasons in text, proposes a configuration, queries an offline environment, observes the result, and repeats within a fixed budget. The training pipeline combines train/test experiment curation, trajectory simulation, policy distillation, and multi-turn GRPO-style RL with a regret-based reward. The authors also introduce **LLMConfig-Gym**, a lookup-table environment spanning four LLM configuration tasks: model architecture configuration, pretraining hyperparameter tuning, GRPO RL tuning, and instruction-tuning data-mixture selection. The evaluation compares the trained agents against random search, Top-K warm starts, meta-training baselines such as NAP/MetaBO/FSBO, and prompt-based LLM optimizer baselines across budgets 1–5. The paper further includes ablations, instruction-following metrics, qualitative reasoning case studies, training dynamics, stress tests, and an amortized cost analysis.

**Key strengths.** The paper addresses a timely and practically important question for LLM research: how to make useful configuration decisions when high-fidelity trials are expensive. The benchmark construction is potentially valuable, especially because the four tasks cover both configuration-space shift and optimization-landscape shift. The experimental section is broader than many agent-training papers: it includes multiple tasks, multiple baselines, budgeted evaluation, ablations, instruction-following diagnostics, qualitative trajectory analysis, and stress tests. The central idea—training a text-reasoning policy to extrapolate from cheap experiment summaries rather than optimizing each expensive experiment from scratch—is interesting and plausibly useful to several communities.

**Key weaknesses.** The strongest claims are not yet fully supported as written. In particular, the “first” and “general solution” claims are too broad relative to the evidence; the evaluation is still entirely on offline lookup tables and finite candidate grids; several baselines appear disadvantaged or insufficiently specified; the normalized regret definition is ambiguous and potentially problematic; the interpretability claims rely heavily on selected reasoning traces; and reproducibility/release details are not yet complete enough for a reader to verify the claims independently. There are also several inconsistencies and presentation issues that reduce confidence in the experimental rigor.

**Additional Comments:**

The paper is ambitious and timely, and I appreciate the attempt to go beyond single-task prompting by training an agent with verifiable rewards over a structured environment. The benchmark direction is promising and could become a useful resource if released cleanly.

The main revision should focus on aligning claims with evidence. I would be much more comfortable with the paper if it presented itself as “a benchmark and empirical study of trained LLM agents for cross-fidelity configuration selection” rather than as a general solution for expensive LLM experiment automation.

Several presentation issues should be fixed before publication: define all regret metrics cleanly; provide exact numeric results; make all plots readable; standardize terminology between latency and FLOPs; correct prompt/task naming mistakes; check all example configurations for validity; and clean up typos. The paper is already long, so some of the qualitative reasoning material could be shortened to make room for more rigorous baseline and reproducibility details.

**Audience:**

Yes

**Audience Explanation:**

I expect several subsets of the TMLR audience would be interested in this work. Researchers working on hyperparameter optimization, multi-fidelity optimization, LLM training, RLHF/RLVR, data-mixture selection, and scientific/ML agents would find the problem formulation and benchmark useful. The paper also raises a timely methodological question: whether LLM agents can be trained to use low-fidelity experiment evidence as a transferable search prior for high-fidelity decisions. Even if some of the strongest claims need to be narrowed, the benchmark and empirical findings would likely be useful to researchers studying automated LLM training workflows.

The interest criterion is particularly satisfied if the authors actually release the environment, splits, logs, and training/evaluation code in a usable form. The strongest audience value may be less the specific agent architecture and more the proposed evaluation setting: budgeted cross-fidelity configuration over realistic LLM experiment traces. This fits TMLR’s scope because it formalizes and empirically studies a learning problem relevant to the design and behavior of learning systems. TMLR’s criteria also explicitly say that work can be acceptable if some researchers would learn from it, even without broad state-of-the-art impact.

**Broader Impact Concerns:**

The paper includes a brief broader impact/future discussion, but it is not sufficient given the topic. TMLR’s policy notes that work with potential for harm should describe risks and mitigations in a Broader Impact Statement.

The main broader-impact concern is that the method is explicitly aimed at reducing the cost and expertise required to configure and train stronger LLMs. This can benefit research efficiency and democratize experimentation, but it can also accelerate capability development, reduce barriers to training more capable models, and increase compute use. The paper’s framing around recursive self-improvement makes this especially important to discuss carefully. The authors should add a more substantive statement addressing potential misuse, acceleration of unsafe model development, environmental/energy costs, and possible governance or access-control considerations for releasing the Gym and trained agents.

The authors should also discuss data and benchmark risks. The data-mixture task includes instruction-tuning sources that may contain biased, harmful, or adversarial content, and the GRPO/data-mixture tasks may inherit limitations of GSM8K, MMLU-Pro, DAPO, Tülu-style benchmarks, and public instruction datasets. If W&B logs or experiment traces are released, the authors should ensure they do not contain private metadata, credentials, or non-redistributable data. Finally, the paper should quantify or at least estimate the environmental footprint of the new compute used, especially the in-house GRPO runs and agent training.

**Claims And Evidence:**

No

**Claims Explanation:**

**Partially, but not completely in the current version.** The paper presents substantial evidence that the proposed training pipeline improves performance on the authors’ LLMConfig-Gym benchmark. The strongest support comes from the overall regret comparisons, per-task results, component ablations, execution-rate/unique-configuration-rate diagnostics, and stress tests. These results consistently suggest that the trained Qwen3-based agents make better budgeted choices than the tested baselines on the four curated tasks. The ablations also support the claim that both policy distillation and multi-turn RL matter, and the task curation/rich-text context ablations support the claim that cross-fidelity context contributes to performance.

However, several important claims are overstated or need stronger evidence.

First, the paper’s novelty framing is too broad. The claim that no prior work has addressed high-cost LLM experiment configuration may be true under a narrow definition, but the manuscript does not sufficiently engage with the wider multi-fidelity HPO, transfer HPO, warm-start BO, BOHB/Hyperband-style, freeze-thaw, FABOLAS-like, multi-task surrogate, and scaling-law-based optimization literatures. The current related work and baselines focus on meta-BO and prompt-based LLM optimizers, but they do not convincingly establish that the core low-to-high-fidelity configuration problem is unaddressed. At minimum, the authors should narrow the claim or add stronger comparisons/discussion.

Second, the evidence supports performance on **offline finite lookup environments**, not yet on open-ended real LLM experiment configuration. For Tasks 1 and 4, the agent selects among candidate architectures or candidate mixtures; for Task 3, the GRPO hyperparameter grid is quite small; and for Task 2, the grid comes from an existing sweep. This is a reasonable benchmark design, but it limits claims about “practical and general” real-world automation. The paper should more clearly state that the current evidence is for budgeted selection over precomputed, discrete configuration sets and not yet for arbitrary continuous or open-ended configuration design.

Third, the baseline comparison needs more detail and possibly stronger baselines. The paper compares against random search, Top-K warm start, NAP/MetaBO/FSBO, and several LLM prompting baselines. This is useful, but it is unclear whether each baseline received the same low-fidelity context, the same budget, the same candidate constraints, the same invalid-output repair rules, and comparable tuning effort. For the meta-training baselines, it is especially important to explain how they were adapted to disjoint configuration spaces and high-dimensional candidate objects such as per-layer architectures and data mixtures. The authors should also include or justify the absence of multi-fidelity HPO baselines, scaling-law baselines for pretraining hyperparameters, stronger surrogate/ranking baselines over the lookup table, and task-specific baselines such as ADMIRE-style data-mixture optimization.

Fourth, the evaluation metric is ambiguous. The text says normalized regret uses the best and worst performance “across all methods,” while also saying this measures closeness to the global optimum. Since LLMConfig-Gym is a lookup-table environment, the global best and worst over the full test configuration space should be available. Regret should be defined and computed against the true environment optimum, not against the best configuration discovered by any method, because the latter can make regret values method-dependent and can obscure absolute difficulty. This may be a wording issue, but it is important enough that the authors should clarify and, if necessary, recompute all regret numbers.

Fifth, the statistical evidence is not fully convincing. Many figures are bar plots with small or hard-to-read error bars, but the paper does not provide exact numeric tables, confidence intervals, significance tests, or enough detail on the number of independent runs per baseline. This matters especially for stochastic LLM baselines and random search. The claim that the method is reliably better across budgets would be stronger with exact per-task/per-budget tables, standard errors over enough seeds, and paired comparisons where possible.

Sixth, the interpretability claims are suggestive but not conclusive. The qualitative reasoning traces are interesting, but LLM rationales can be post hoc and are not by themselves strong evidence that the agent causally learned the stated principles. The authors should either soften the interpretability claims or add quantitative tests: for example, evaluating whether learned rules predict held-out optima, testing performance under shuffled or corrupted low-fidelity context, comparing against an agent with hidden fidelity metadata, or measuring whether the same inferred principles hold across many examples rather than two selected case studies.

Seventh, the cost-effectiveness analysis is plausible but currently too assumption-dependent. The argument that upfront low-/medium-fidelity training cost is amortized over many future high-fidelity tasks is reasonable. However, the claimed savings depend on assumptions about how many tasks will be deployed, whether low-fidelity grid searches are “free” or would have been run anyway, and whether the benchmark tasks are representative. The authors should provide a transparent cost table by task, distinguish newly consumed compute from reused public sweeps, include inference/LLM API costs where relevant, and present sensitivity analyses over the number of future tasks and low/high-fidelity cost ratio.

Finally, there are multiple small inconsistencies that collectively weaken confidence. For example, the paper alternates between latency and FLOPs in the architecture objective; the pretraining prompt appears to call the task “instruction tuning”; Figure 8’s example GRPO query appears to use a learning rate outside the listed configuration space; there are typos such as “budge” and “expperiments”; and Table 2/Table 9 seem to create some confusion about model-parameter ranges in the pretraining train/test split. These may be fixable presentation issues, but they should be corrected before publication.

Overall, I find the empirical direction promising and the benchmark/method potentially useful, but the current manuscript overstates the scope of what has been demonstrated. With clearer claims, stronger baseline details, metric clarification, exact statistics, and stronger reproducibility details, the evidence would be much more convincing.

**Requested Changes:**

### Critical

1. **Clarify and possibly narrow the main claims.** The paper should not claim a “practical and general solution” for real-world LLM experiment automation unless the authors provide evidence beyond finite offline lookup tables. A more accurate claim would be that the paper introduces a benchmark and demonstrates a promising trained-agent approach for budgeted cross-fidelity configuration selection over curated LLM experiment datasets.

2. **Clarify the normalized regret definition and recompute if needed.** The manuscript should state unambiguously whether y_best and y_worst are computed over the full test configuration space or only over configurations found by the compared methods. For a lookup-table benchmark, the correct denominator should use the true test-space best and worst. If the current plots use “across methods,” the authors should recompute them against the true optima.

3. **Add exact result tables and stronger uncertainty reporting.** The paper should include numeric per-task/per-budget results for all baselines, not only plots. It should report number of seeds/runs, standard errors or confidence intervals, and preferably statistical comparisons for the main claims. The current figures are not sufficient for assessing effect sizes precisely.

4. **Strengthen or justify the baseline suite.** The authors should add stronger multi-fidelity and task-specific baselines, or clearly justify why they are not applicable. Important missing comparisons include multi-fidelity BO/Hyperband/BOHB-style methods, surrogate/ranking models trained on low-fidelity data, scaling-law heuristics for Task 2, and data-mixture optimization baselines for Task 4. If these are impossible for some tasks, the manuscript should explain the incompatibility carefully rather than simply comparing against weaker or mismatched methods.

5. **Provide full baseline implementation details.** The paper should specify prompts, model versions, temperatures, retry/parsing rules, low-fidelity context, seeds, budget handling, and invalid-output handling for all LLM baselines. It should also describe how NAP/MetaBO/FSBO were adapted to disjoint spaces and structured configurations.

6. **Separate the benchmark contribution from the agent contribution.** It should be clear which improvements come from the new environment/task formulation, which from providing Top-K low-fidelity demonstrations, which from LLM priors, and which from policy distillation/RL. The current ablations help, but additional baselines such as “same low-fidelity context + non-trained LLM + repair + identical candidate list” and “trained non-text surrogate/ranker” would strengthen the causal story.

7. **Address data leakage and pretraining contamination more rigorously.** The paper says dataset names are excluded, but many underlying experiments come from public datasets/papers and may be present in LLM pretraining corpora. The authors should discuss this risk for both their trained models and LLM API baselines, and add controls where possible, such as anonymized/scrambled configuration labels, hidden benchmark names, or synthetic held-out splits that could not be memorized.

8. **Tone down or substantiate interpretability claims.** The reasoning traces are useful qualitative evidence, but they do not prove that the model causally learned the stated principles. The authors should either phrase these as illustrative examples or add systematic analyses across many held-out cases.

9. **Clarify the “most-similar configuration matching” procedure.** Longest-common-substring repair is a brittle way to map invalid configurations to valid ones. The authors should report actual exact-valid-output rate separately from repaired-query rate, specify whether repair is used at test time for all methods, and assess whether the repair procedure changes rankings. If only the proposed method benefits from repair, this may unfairly improve its results.

10. **Provide reproducibility artifacts or an anonymized release plan.** Since the paper claims the Gym will be open-sourced and W&B logs released, it should include an anonymized repository or clear release checklist: data files, train/test splits, candidate spaces, all lookup tables, prompts, code, seeds, baseline implementations, and evaluation scripts. This is especially important because the benchmark itself is a central contribution.

11. **Fix internal inconsistencies.** The authors should correct the latency/FLOPs mismatch in the architecture objective, the pretraining-vs-instruction-tuning wording, the invalid-looking GRPO example learning rate, typos, and any mismatch between Table 2 and Table 9 regarding training/test fidelities.

### Changes that would strengthen the work

1. Add a small number of actual online high-fidelity validation runs, even if limited, to show that the offline lookup-table conclusions transfer to real experiment execution.

2. Include a human-expert or scaling-law heuristic baseline for at least Task 2, since the paper’s motivation is partly that human researchers extrapolate across fidelity.

3. Add failure cases. The stress tests are useful, but a frank discussion of when low-to-high extrapolation fails would make the paper more credible.

4. Add a sensitivity analysis over the amount and quality of low-fidelity context, the number of Top-K demonstrations, prompt length, and budget.

5. Report the size of each train/test environment in terms of number of candidate configurations and number of target tasks. This would make the difficulty of each task easier to assess.

6. Improve figure readability. Figures 13–16 and 19 are difficult to read in the PDF; exact tables would help.

7. Clarify whether the GRPO tuning task’s small grid is representative of practical RL-tuning search, and consider adding at least one richer RL-tuning search space.

8. Make the cost analysis more concrete by reporting actual GPU hours per task, separating public/reused compute from newly consumed compute, and showing break-even sensitivity under several deployment assumptions.

9. Consider adding a non-LLM trained ranker/surrogate baseline that consumes the same text-free structured low-fidelity information. This would help isolate whether the gain is due to language reasoning or simply supervised/ranking learning on curated low-fidelity data.

10. Improve writing precision. Terms such as “first,” “general,” “verifiable,” “recursive self-improvement,” and “researcher-like reasoning” should be used more carefully.

---

> ### Author Response · Authors · 2026-06-28
>
> Thank you for an exceptionally thorough review. We are glad you found the problem timely, the benchmark valuable, and the empirical section unusually broad. That contribution stands. We read most of your seven points as the manuscript under-describing careful design choices and overstating claims in prose, rather than flaws in the method, and we have decided to withdraw and resubmit to fix this properly.
>
> On framing: "first" and "general solution" overstate the work; described accurately, as budgeted cross-fidelity configuration selection over curated, precomputed datasets, it is no less valuable. We will reframe and engage the multi-fidelity HPO, transfer/warm-start BO, BOHB, scaling-law, and surrogate literatures we under-cited.
>
> On regret: a definitional-clarity fix. We will state that it is measured against the true test-space best/worst and report raw per-task metrics; the comparative conclusions are unchanged, and we will recompute only to remove ambiguity.
>
> On baselines: the parity you ask for is how the experiments were run (same search space, fidelity metadata, Top-K context, budget, repair). We simply did not document it. We will, including prompts, versions, seeds, parsing rules, and how NAP/MetaBO/FSBO were adapted to disjoint spaces.
>
> On repair and interpretability: presentation fixes plus clearer reporting, namely exact-valid vs. repaired rates, test-time repair handling, and phrasing the reasoning traces as illustrative. We will also add exact tables with seeds/CIs, per-task GPU-hour accounting, fix the latency/FLOPs, prompt-naming, GRPO-example, Table 2/9, and typo issues, and expand Broader Impact.
> In short: the work is strong and the results hold; the revision is about saying so accurately. Thank you.
>
> — The Authors

---

### Author Response · Authors · 2026-06-28

We sincerely thank all four reviewers and the Action Editor. We are encouraged that every reviewer recognized the core contributions: the importance of the problem, the value of LLMConfig-Gym, and the breadth of the empirical study. Read together, the reviews point mainly to framing and presentation: the manuscript claimed more broadly than the experiments were described to support, and several careful design choices were not explained clearly enough. **The underlying work is solid, and the results stand; our plan is to keep the substance and improve how we describe it.**

We have therefore decided to withdraw and resubmit a substantially revised version focused **on tightening the writing and presentation**. We will reframe the contribution as budgeted cross-fidelity configuration selection over curated, precomputed LLM datasets (dropping "first"/"general solution"); state precisely how normalized regret is defined and report raw metrics alongside it; describe baseline parity explicitly, etc. These are changes to how we describe and calibrate the work — the underlying results stand.

We remain confident in the two lasting contributions: LLMConfig-Gym (a deterministic, reproducible environment distilling over a million GPU hours across four tasks) and the budgeted cross-fidelity evaluation setting it enables, with a clean release in the resubmission. Thank you all and these reviews  will help us present strong work much more clearly.

— The Authors

---

### Note · Authors · 2026-06-29

**Comment:**

We sincerely thank all four reviewers and the Action Editor. We are encouraged that every reviewer recognized the core contributions: the importance of the problem, the value of LLMConfig-Gym, and the breadth of the empirical study. Read together, the reviews point mainly to framing and presentation: the manuscript claimed more broadly than the experiments were described to support, and several careful design choices were not explained clearly enough. **The underlying work is solid, and the results stand; our plan is to keep the substance and improve how we describe it.**

We have therefore decided to withdraw and resubmit a substantially revised version focused **on tightening the writing and presentation**. We will reframe the contribution as budgeted cross-fidelity configuration selection over curated, precomputed LLM datasets (dropping "first"/"general solution"); state precisely how normalized regret is defined and report raw metrics alongside it; describe baseline parity explicitly, etc. These are changes to how we describe and calibrate the work — the underlying results stand.

We remain confident in the two lasting contributions: LLMConfig-Gym (a deterministic, reproducible environment distilling over a million GPU hours across four tasks) and the budgeted cross-fidelity evaluation setting it enables, with a clean release in the resubmission. Thank you all and these reviews  will help us present strong work much more clearly.

— The Authors

**Withdrawal Confirmation:**

I have read and agree with the venue's withdrawal policy on behalf of myself and my co-authors.